# Runtime Safety through Adaptive Shielding: From Hidden Parameter Inference to Provable Guarantees

## Abstract

Unseen shifts in environment dynamics, driven by hidden parameters such as friction or gravity, can trigger safety risks during deployment. We develop a runtime shielding mechanism for reinforcement learning, building on the formalism of constrained hidden-parameter Markov decision processes. Function encoders enable real-time inference of hidden parameters from observations, allowing the shield and the underlying policy to adapt online. To further promote safe policy learning, we introduce a safety-regularized objective that augments reward maximization with a bounded safety measure. This objective encourages the selection of actions that minimize long-term safety violations. The shield constrains the action space by forecasting future safety risks (such as obstacle proximity) and accounts for uncertainty via conformal prediction. We prove that the proposed mechanism satisfies probabilistic safety guarantees and yields optimal policies within safety-compliant policies. Experiments across diverse environments with varying hidden parameters show that our approach reduces safety violations while maintaining effective task-solving performance, and achieving robust out-of-distribution generalization.

## 1 Introduction

Robots and other autonomous systems must operate safely in open-world environments where the underlying dynamics can vary due to hidden parameters such as mass distribution, friction, or terrain compliance. These parameters often change across episodes and remain unobserved, introducing safety risks and challenging the generalization capabilities of reinforcement learning (RL) systems (Kirk et al., 2023; Benjamins et al., 2023). Ensuring robust and safe behavior under such uncertainty is essential in domains like autonomous driving and robotic manipulation, where failures can have serious real-world consequences.

Despite recent progress in hidden parameter-aware and safe RL, existing methods often trade off adaptability and safety. Approaches such as hypernetworks, contextual models, or mixtures-of-experts (Rezaei-Shoshtari et al., 2023; Beukman et al., 2023; Celik et al., 2024) demonstrate strong adaptation to varying dynamics, but typically lack explicit mechanisms for guaranteeing safety under uncertainty. Conversely, safe reinforcement learning (RL) frameworks based on constrained Markov decision processes (CMDPs) (Achiam et al., 2017; Tessler et al., 2018; Wachi & Sui, 2020; Yang et al., 2020; 2022) enforce safety by imposing constraints on cumulative costs. Methods such as state augmentation (Li et al., 2022) and shielding (Alshiekh et al., 2017; Yang et al., 2023) improve safety while maintaining compatibility with a wide range of safe RL algorithms. However, these approaches typically assume stationary dynamics and lack the ability to adapt in real time to hidden parameter shifts.

To address this gap, we propose a runtime shielding framework for reinforcement learning that adapts online to hidden parameters while offering provable probabilistic safety guarantees. Central to our approach is the use of function encoders (Ingebrand et al., 2024b; 2025), a compact and expressive model class that infers environment dynamics from transition data by projecting them onto neural basis functions. This representation enables fast, online adaptation of both the policy and shield without retraining.

To ensure safe learning and adaptation, our approach combines two complementary mechanisms that operate proactively during training and reactively at execution. First, we introduce a safety-regularized objective that augments rewards with a cost-sensitive value estimate, encouraging the policy to avoid unsafe behavior during training. However, this objective alone cannot guarantee safety, particularly under distribution shift. To address this, we augment policy execution with an adaptive shield that samples candidate actions from the policy, predicts future states using a function encoder, and applies conformal prediction to quantify uncertainty in these forecasts. Actions that fail to meet a safety margin are filtered out, ensuring that only safe actions are executed. Empirical evaluations in Safe-Gym benchmarks (Ji et al., 2023), including out-of-distribution scenarios with unseen hidden parameters, demonstrate that our method reduces safety violations compared to baselines, achieving robust generalization with minimal runtime overhead.

In summary, our main contributions are:

- **Safety-Regularized RL Objective**: We propose a new objective that balances reward and safety by integrating a cost-sensitive value function, and encouraging a low-violation behavior policy.
- **Online Hidden-Parameter Adaptation**: We leverage function encoders to infer hidden parameters from transitions, enabling efficient policy and shield adaptation without retraining.
- **Adaptive Shield with Probabilistic Guarantees**: We develop an adaptive, uncertainty-aware runtime shield that filters unsafe actions using conformal prediction, ensuring safety during execution with provable probabilistic guarantees.

## 1.1 RELATED WORK

**Safe Reinforcement Learning.** Safe RL methods often employ constrained MDP formulations to ensure compliance with safety constraints. Constrained policy optimization (CPO) remains foundational, effectively balancing performance and safety (Achiam et al., 2017; Wachi & Sui, 2020). Further techniques use learned recovery policy to ensure safe action execution (Thananjeyan et al., 2020). Recent zero-violation policy methods in RL aim to minimize safety violations using techniques like genetic cost function search, energy-based action filtering, primal-dual, and primal algorithms (Hu et al., 2023; Zhao et al., 2021; Ma et al., 2024; Liu et al., 2021; Bai et al., 2023). However, these approaches often face scalability issues, rely on restrictive assumptions, or are limited to simple environments. Unlike these approaches, we introduce the safety regularized-objective that can be integrated into the optimization process of any CMDP-based RL algorithms. Shielding frameworks proactively filter unsafe actions, selectively sampling safe actions (Alshiekh et al., 2017; Carr et al., 2023; Yang et al., 2023). Recent developments on shielding integrate adaptive conformal prediction into safety frameworks, enhancing uncertainty quantification for safety-critical planning (Sheng et al., 2024a;b). Control barrier functions (CBFs) offer an alternative certificate-based safety mechanism. However, learning a valid barrier certificate is often difficult under uncertain or varying dynamics, as it typically requires explicit model knowledge or robust bounds on the hidden parameters (Choi et al., 2020; Cheng et al., 2023; Ganai et al., 2023; Wang et al., 2023; Xiao et al., 2023). For details on how this connects to our adaptive shielding mechanism, see Appendix E. However, unlike existing methods, which are not designed to address varying hidden dynamics, our approach concurrently enhances safety through a safety-regularized objective and adaptive shielding while adapting to dynamic hidden parameters using function encoders.

**Contextual or Hidden-Parameter Reinforcement Learning.** Hidden parameters, often termed context, have been studied in recent context-aware reinforcement learning approaches, demonstrating their importance for generalization (Benjamins et al., 2023). When algorithms are provided with knowledge of the hidden parameters, they are often directly integrated into the model. For example, contextual recurrent state-space models explicitly incorporate known contextual information to enable zero-shot generalization (Prasanna et al., 2024). Contextualized constrained MDPs further integrate context-awareness into safety-prioritizing curricular learning (Koprulu et al., 2025). A common approach to handle unknown context information is to infer it from observational history using transformer models (Chen et al., 2021). Hypernetwork-based methods utilize adapter modules to adjust policy networks based on inferred contexts (Beukman et al., 2023). Mixture-of-experts architectures leverage specialized experts, using energy-based models to handle unknown contexts probabilistically (Celik et al., 2024). However, these works primarily focus on enhancing generalization to varying dynamics without incorporating safety mechanisms during adaptation in contrast to our method.

**Generalization in Reinforcement Learning.** Generalization in RL, including zero-shot transfer and meta learning, is crucial for robust policy adaptation to varying dynamics. For example, meta-learning approaches, such as MAML (Finn et al., 2017), allow rapid parameter adaptation from minimal interaction data. Safe meta RL (Khattar et al., 2023; Guan et al., 2024) extends meta-reinforcement learning to adapt to new tasks while adhering to safety constraints. However, meta-learning approaches involve parameter updates during adaptation, whereas our framework focuses on rapid, online inference of hidden parameters without requiring such updates. Hypernetwork-based zero-shot transfer methods explicitly condition policies on task parameters (Rezaei-Shoshtari et al., 2023). Function encoders, i.e. neural network basis functions, have demonstrated strong zero-shot transfer by using the coefficients of the basis functions as a fully-informative, linear representation of the dynamics (Ingebrand et al., 2024b;a). Single-episode policy transfer and adaptive methods effectively handle environment changes by encoding historical context (Yang et al., 2019; Chen et al., 2022). Advanced context encoder designs further improve robustness and fast adaptation capabilities (Luo et al., 2022). While these methods excel at adapting to varying dynamics, they do not address safety constraints during adaptation, leaving agents vulnerable to unsafe actions in unseen environments.

## 2    PROBLEM FORMULATION

Constrained hidden-parameter MDPs (CHiP-MDPs) model environments with varying transition dynamics, where a cost function is introduced alongside a reward function to address safety constraints. A CHiP-MDP extends the HiP-MDP framework (Konidaris & Doshi-Velez, 2014) and is defined by the tuple $\mathcal{M} = (S, A, \Phi, T, R, C, \gamma, P_\Phi)$, where $S$ and $A$ are the state and action spaces, $R : S \times A \times S \to \mathbb{R}$ is a reward function, $C : S \times A \times S \to [0, 1]$ is a cost function, and $\gamma \in (0, 1)$ is the discount factor. The transition dynamics $T : S \times A \times \Phi \to S$ depend on a hidden parameter $\phi \in \Phi$. For a specified hidden parameter $\phi \in \Phi$, we denote the transition dynamics as $T_\phi : S \times A \to S$. The prior $P_\Phi(\phi)$ over the parameter space $\Phi$ represents the distribution of these hidden parameters. We denote the initial state distribution as $\mu_0$.

Since the hidden parameters $\phi$ are unknown to the agent, it must infer changes in the environment dynamics from observations. To this end, the agent follows a policy $\pi : S \times \mathcal{B} \to A$, where $\mathcal{B}$ denotes the set of learned representations of the transition dynamics $T_\phi$. We denote the resulting representation by $b_\phi$ for each $\phi$. The objective of the agent is to maximize expected cumulative discounted reward while satisfying safety constraints in a CHiP-MDP $\mathcal{M}$. To formalize this objective, we define the reward action-value function, for a parameter $\phi$, as:

$$Q_R^\pi(s, a, b_\phi) = \mathbb{E}_{\pi, T_\phi} \left[ \sum_{t=0}^\infty \gamma^t R(s_t, a_t, s_{t+1}) \mid s_0 = s, a_0 = a, \phi \right]. \tag{1}$$

The corresponding reward state-value function, which averages $Q_R^\pi$ over actions, is:

$$V_R^\pi(s, b_\phi) = \mathbb{E}_{a \sim \pi(\cdot|s, b_\phi)} \left[ Q_R^\pi(s, a, b_\phi) \right]. \tag{2}$$

Finally, the reward objective is defined as :

$$J_R(\pi) = \mathbb{E}_{\phi \sim P_\Phi, s_0 \sim \mu_0(\cdot|\phi), a_0 \sim \pi(\cdot|s_0, b_\phi)} \left[ Q_R^\pi(s_0, a_0, b_\phi) \right]. \tag{3}$$

Likewise, the cost objective is defined the same way, replacing the reward function $R$ with the cost function $C$: $J_C(\pi) = \mathbb{E}_{\phi \sim P_\Phi, s_0 \sim \mu_0(\cdot|\phi), a_0 \sim \pi(\cdot|s_0, b_\phi)} \left[ Q_C^\pi(s_0, a_0, b_\phi) \right]$. The safety constraints aim to minimize the average cost rate. To this end, we state our problem below.

**Problem.** Given a CHiP-MDPs $\mathcal{M} = (S, A, \Phi, T, R, C, \gamma, P_\Phi)$ where the transition dynamics $T_\phi$ are fully unknown and vary with a hidden parameter $\phi$, find an optimal policy $\pi^*$ that maximizes the expected cumulative discounted reward $J_R(\pi^*)$ while satisfying the safety constraints on the average cost rate,

$$\xi^{\pi^*}(s, \phi) = \lim_{H \to \infty} \frac{1}{H} \mathbb{E}_{\pi^*, T_\phi} \left[ \sum_{t=0}^{H-1} C(s_t, a_t, s_{t+1}) \mid s_0 = s, \phi \right] \le \delta, \tag{4}$$

where $\delta \in (0, 1)$ is a failure probability. Note that the transition dynamics depend on the hidden parameter $\phi$, but the policy depends on a representation of the hidden parameter, $b_\phi$, derived from any previously observed transitions by $T_\phi$.

To enforce the safety constraint, we define the cost action-value function $Q_C^\pi$ and cost state-value function $V_C^\pi$ by replacing the reward function $R$ with the cost function $C$ from Equations 1 and 2. Minimizing the average cost rate can be achieved by minimizing the cost-value function $V_C^\pi$ (see Appendix N).

# 3 BACKGROUND

We introduce key concepts essential for understanding our methods. First, function encoders have demonstrated robust performance in estimating varying underlying dynamics (Ingebrand et al., 2024b;a; 2025). Second, conformal prediction provides a rigorous framework for quantifying uncertainty (Vovk et al., 2005; Tibshirani et al., 2019; Gibbs & Candès, 2024).

**Function Encoder.** A function encoder (FE) offers a compact and computationally efficient framework for representing functions in terms of neural network basis functions. Consider a set of functions $\mathcal{F} = \{f \mid f : \mathcal{X} \to \mathbb{R}\}$, where $\mathcal{X} \subset \mathbb{R}^n$ is an input space with finite volume. When $\mathcal{F}$ forms a Hilbert space with the inner product $\langle f, g \rangle = \int_{\mathcal{X}} f(x)g(x)dx$, any $f \in \mathcal{F}$ can be expressed using a basis $\{g_1, g_2, \ldots, g_k\}$ as $f(x) = \sum_{i=1}^{k} b_i g_i(x)$, where $b_i$ are unique coefficients. To determine the coefficients, we solve the following least-squares optimization problem:

$$(b_1, b_2, \cdots, b_k) := \underset{(b_1, b_2, \cdots, b_k) \in \mathbb{R}^k}{\arg\min} \left\| f - \sum_{j=1}^{k} b_j g_j \right\|_2^2. \tag{5}$$

For more information on how to train the neural network basis functions, see Ingebrand et al. (2025).

**Conformal Prediction.** Conformal Prediction (CP) allows for the construction of prediction intervals (or regions) that are guaranteed to cover the true outcome with a user-specified probability, under minimal assumptions. For exchangeable random variables $\{Z_i\}_{i=1}^{t+1}$, CP constructs a region satisfying: $\mathbb{P}(Z_{t+1} \le \Gamma_t) \ge 1 - \delta$, where $\delta \in (0, 1)$ is the failure probability, and the threshold $\Gamma_t = Z_{(q)}$ is the $q$-th order statistic of $\{Z_1, \ldots, Z_t\}$, with $q = \lceil (t+1)(1-\delta) \rceil$. Adaptive Conformal Prediction (ACP) extends this to non-stationary settings by making the threshold learnable. For more information on conformal prediction, see Shafer & Vovk (2008); Gibbs & Candès (2021).

# 4 APPROACH

Our approach has three main components. First, we introduce a novel *safety-regularized objective*. This objective is used during optimization and encourages the policy to converge toward a zero-violation policy. Second, we use a function encoder to represent underlying dynamics $T_\phi$, enabling *online adaptation*. Finally, we leverage this dynamics representation to construct an *adaptive shield*. The shield adjusts safe regions by conformal prediction and blocks unsafe actions online.

## 4.1 SAFETY-REGULARIZED OBJECTIVE

To promote safe policy learning, we introduce a safety measure, $Q_{\text{safe}}^\pi(s, a, b_\phi)$, which quantifies the safety of an action $a \sim \pi(\cdot|s, b_\phi)$, given the learned representation $b_\phi \in \mathbb{R}^k$. Higher values of $Q_{\text{safe}}^\pi$ indicate actions with lower long-term costs under policy $\pi$. Since we aim to minimize the cost action-value function $Q_C^\pi$, higher $Q_{\text{safe}}^\pi$ values correspond to lower $Q_C^\pi$ values. Based on this intuition, we define $Q_{\text{safe}}^\pi(s, a, \phi)$ for an action $a \sim \pi(\cdot|s, b_\phi)$ as:

$$Q_{\text{safe}}^\pi(s, a, b_\phi) = -\frac{\int_{B(a,\epsilon) \cap A} \pi(x \mid s, b_\phi) Q_C^\pi(s, x, b_\phi)dx}{V_C^\pi(s, b_\phi) + \epsilon} \tag{6}$$

where $\epsilon > 0$ is a small constant ensuring numerical stability, and $B(a, \epsilon)$ denotes a small ball of radius $\epsilon$ centered at $a$.

This formulation bounds the value in $(-1, 0]$ by its design. For continuous action spaces, the probability of taking a specific action is always 0, so we integrate the value over a small interval including that action. For practical implementation, we use a Monte Carlo approximation to the integral by sampling several values around an action $a$ and then aggregating them.

A value near 0 for $Q_{\text{safe}}^\pi(s, a, b_\phi)$ indicates one of two scenarios: 1) Safety: where the policy selects an action $a$ resulting in near-zero long-term cost violations, i.e., $Q_C^\pi(s, a, b_\phi) \approx 0$; or 2) Exploration: where the probability of selecting action $a$ is small, i.e., $\pi(a|s, b_\phi) \approx 0$. In contrast, when $Q_{\text{safe}}^\pi(s, a, b_\phi)$ is near $-1$, it indicates that actions around $a$ substantially contributes to the expected cumulative cost $V_C^\pi(s, b_\phi)$, posing a higher risk compared to other action choices at state $s$. See Appendix K for details on the design choice.

To integrate this safety measure into policy optimization, we define an augmented action-value function, $Q_{\text{aug}}^\pi(s, a, b_\phi) = Q_R^\pi(s, a, b_\phi) + \alpha Q_{\text{safe}}^\pi(s, a, b_\phi)$, where $Q_R^\pi(s, a, b_\phi)$ is the reward action-value function, and $\alpha \geq 0$ is a hyperparameter balancing safety and reward. Our *safety-regularized objective* (SRO) is:

$$J_{\text{aug}}(\pi) = \mathbb{E}_{\phi \sim P_\phi, s_0 \sim \mu_0(\cdot|\phi), a_0 \sim \pi(\cdot|s_0, b_\phi)} \left[ Q_{\text{aug}}^\pi(s_0, a_0, b_\phi) \right]. \tag{7}$$

A larger $\alpha$ encourages the policy to prioritize safe actions that result in zero-violation costs or to select under-explored actions with lower assigned probabilities. Next, we introduce a proposition which justifies this choice of objective.

**Proposition 1.** Let $\Pi_{\text{zero-violation}}$ denote the set of zero-violation policies, defined as $\{\pi \mid J_C(\pi) = 0\}$. Then, for any $\alpha \geq 0$, the optimal policy obtained by maximizing the safety-regularized objective function $J_{\text{aug}}(\pi)$ within $\Pi_{\text{zero-violation}}$ is equivalent to the optimal policy obtained by maximizing the standard reward objective $J_R(\pi)$ within the same set of policies.

*Proof Sketch.* For any policy within the zero-violation set, all actions sampled from the policy lead to $Q_C^\pi(s, a, \phi) = 0$. By our design of the safety term, this condition implies $Q_{\text{safe}}^\pi(s, a, b_\phi) = 0$. Substituting this into our regularized objective, $J_{\text{aug}}(\pi)$ simplifies to $J_R(\pi)$. $\square$

Proposition 1 proves that the safety regularization does not degrade performance unnecessarily when an agent already behaves safely. Specifically, it guarantees that if we focus only on the set of policies that satisfy all safety constraints, maximizing the safety-regularized objective is equivalent to maximizing the standard reward objective. For a theoretical analysis of how the safety-regularized objective (SRO) combines with TRPO and CPO, see Appendix B. The detailed integration of SRO into the actor-critic training loop is provided in Appendix D.

### 4.2 Inferring Hidden Parameters Online

To infer the underlying dynamics $T_\phi$ and predict the next state $s_{t+1}$ based on transition samples and $(s_t, a_t)$, we use a function encoder, denoted by $\hat{f}_{\text{FE}}$. Given observed transition samples $\{(s_i, a_i, s_{i+1})\}_{i=1}^{t-1}$ and the current state-action pair $(s_t, a_t)$, the function encoder predicts the next state $\hat{s}_{t+1}$ as: $\hat{s}_{t+1} = \hat{f}_{\text{FE}}(s_t, a_t) = \sum_{i=1}^k b_i \cdot g_i(s_t, a_t)$, where $g_i(s_t, a_t)$ are pretrained basis functions, and $b_i$ are coefficients derived from a subset of transition samples. These coefficients $b_i$ additionally serve as a representation for $T_\phi$. Due to the properties of basis functions, these representations are fully informative and linear (Ingebrand et al., 2024b). We concatenate them with the state to form an augmented input $(s_t, b_1, \ldots, b_k)$, which the policy uses as input. As the agent interacts with the environment, collecting new transitions $(s_t, a_t, s_{t+1})$, we refine the coefficients $b_i$ by solving Equation 5 with updated transition samples. Consequently, the agent receives an online representation of the dynamics. Note that with a fixed number of basis functions $k$, the computation, involving the inverse of a $k \times k$ matrix, remains efficient even for large samples. We denote the coefficients $(b_1, \cdots, b_k)$ as $b_\phi$ for the dynamics $T_\phi$.

### 4.3 Adaptive Shielding Mechanism

To ensure safety during policy execution, we propose an adaptive shielding mechanism that dynamically intervenes based on uncertainty in model predictions. This shield wraps any underlying policy $\pi$, adjusting actions to prevent unsafe outcomes. We illustrate the shielding process at timestep $t$.

We first introduce the necessary settings. The cost function is defined using an indicator function $\mathbb{I}$ as $C(s_t, a_t, s_{t+1}) = \mathbb{I}\{\nu(e(s_{t+1}), E_{t+1}) \leq 0\}$, where $e : S \to \mathbb{R}^{n_1}$ extracts agent-centric safety features, $E_{t+1} \in \mathbb{R}^{n_2}$ captures environment features, and $\nu : \mathbb{R}^{n_1} \times \mathbb{R}^{n_2} \to \mathbb{R}$ is Lipschitz continuous with Lipschitz constant $L_\nu$. We assume the agent-centric safety features change smoothly,

i.e., $\|e(s_{t+1}) - e(s_t)\| \leq \Delta_{\max}$, where $\Delta_{\max}$ is a bound on the per-step feature change. By the Lipschitz property, the equation $\|e(s_{t+1}) - e(s_t)\| \leq \Delta_{\max}$ implies:

$$\nu(e(s_{t+1}), E_{t+1}) \geq \nu(e(s_t), E_t) - L_\nu \Delta_{\max}. \tag{8}$$

Thus, if $\nu(e(s_t), E_t) > L_\nu \Delta_{\max}$, then $C(s_t, a_t, s_{t+1}) = 0$ for all $a_t \in A$. Since the value $\nu(e(s_t), E_t)$ can be computed at state $s_t$ before selecting action $a_t$ to assess its safety, we call it as the pre-safety indicator.

1. **Pre-Safety Check**: To minimize intervention, we evaluate the pre-safety indicator:

$$\nu(e(s_t), E_t) > L_\nu \Delta, \tag{9}$$

where $\Delta$ is a predefined value larger than $\Delta_{\max}$. If this condition is violated, full safety verification is triggered; otherwise, the policy executes directly. This pre-safety check step improves computational efficiency when full safety verification is excessive.

2. **Action Generation**: The policy $\pi$ generates $N$ candidate actions $\{a_t^{(i)}\}_{i=1}^N$ by sampling from its action distribution $\pi(\cdot \mid s_t, b_\phi)$, where $b_\phi$ derived by a subset of transition samples up to time step $t$ explained in Section 4.2.

3. **Transition Prediction**: For each candidate action $a_t^{(i)}$, a function encoder $\hat{f}_{\text{FE}}$ predicts the next state: $\hat{s}_{t+1}^{(i)} = \hat{f}_{\text{FE}}(s_t, a_t^{(i)})$. Note that any pre-trained forward dynamics model $\hat{f}$ can be used for prediction. However, the function encoder enables inference of varying underlying dynamics and next-state prediction at once.

4. **Safety Verification**: Using ACP, we compute uncertainty-aware safety margins for each action:

$$\text{SafetyScore}(a_t^{(i)}) = \nu\left(e(\hat{s}_{t+1}^{(i)}), \hat{E}_{t+1}\right) - 2L_\nu \Gamma_t, \tag{10}$$

where $\hat{E}_{t+1}$ represents predicted environment features and $\Gamma_t$ is the adaptive conformal prediction bound for $\hat{s}_{t+1}^{(i)}$ and $\hat{E}_{t+1}$ calibrated to maintain a $1 - \delta$ safety probability. Actions are ranked by their safety scores, with positive scores indicating safety compliance.

5. **Action Selection**: Define the safe action set at state $s_t$ as $\hat{A}_{\text{safe}}(s_t) = \{a_t^{(i)} : \text{SafetyScore}(a_t^{(i)}) > 0\}$ and sampled action set as $\hat{A}_{\text{sample}} = \{a_t^{(i)}\}_{i \in [N]}$. The shield executes the following selection rule:

$$a_t^* = \begin{cases} a \sim \mathcal{U}(\text{Top}_k(\hat{A}_{\text{safe}}(s_t))), & \text{if } \hat{A}_{\text{safe}} \neq \emptyset, \\ \arg\max_{a \in \hat{A}_{\text{sample}}} \text{SafetyScore}(a), & \text{otherwise}, \end{cases} \tag{11}$$

where $\mathcal{U}(\text{Top}_k(\cdot))$ denotes a uniform distribution over the top $k$ actions ranked by their safety scores.

When the shield predicts multiple steps $h$ ahead, we repeat the procedure for steps 2, 3, 4, aggregating the safety score over future steps. However, long-term predictions often increase compounding errors and runtime. Thus, we typically use a shorter prediction horizon such as $h = 1$ or $h = 2$.

The following theorem demonstrates that an optimal policy, augmented with an adaptive shield, maximizes the expected cumulative discounted return while maintaining a tight bound on the average cost rate. See Appendix A for details.

**Theorem 1.** Given a Constrained Hidden Parameter MDP $\mathcal{M} = (S, A, \Phi, T, R, C, \gamma, P_\Phi)$ with initial state $s_0 \in S$ and failure probability $\delta \in (0, 1)$, an optimal policy $\pi^* : S \times \Phi \to A$ augmented with an adaptive shield maximizes the expected cumulative discounted return $J_{\text{aug}}(\pi^*)$ while satisfying the average cost rate constraint: for $\phi \sim P_\Phi$ and some $0 \leq \bar{\epsilon} \leq 1$,

$$\xi^{\pi^*}(s, \phi) = \lim_{H \to \infty} \frac{1}{H} \mathbb{E}_{\pi^*, T_\phi} \left[ \sum_{t=0}^{H-1} C(s_t, a_t, s_{t+1}) \mid s_0 = s, \phi \right] \leq \delta + \bar{\epsilon}(1 - \delta). \tag{12}$$

Given this bound, if safe actions exist at each step, this theorem proves that our algorithm achieves a low average cost rate constraint, governed by the ACP failure probability, i.e., $\xi^{\pi^*}(s, \phi) \leq \delta$.

*Proof Sketch.* The ACP provides a probabilistic guarantee on the deviation between the predicted state $\hat{s}_{t+1} = \hat{f}(s_t, a_t)$ and the true state $s_{t+1}$: $(\|s_{t+1} - \hat{s}_{t+1}\| \leq \Gamma_t) \geq 1 - \delta$, where $\Gamma_t$ is the confidence region at time $t+1$. Since $\nu$ is Lipschitz continuous with constant $L_\nu$, we bound the difference in the safety margin between the true and predicted states: $\nu(e(s_{t+1}), E_{t+1}) \geq \nu(e(\hat{s}_{t+1}), \hat{E}_{t+1}) - L_\nu \|e(s_{t+1}) - e(\hat{s}_{t+1})\| - L_\nu \|E_{t+1} - \hat{E}_{t+1}\|$. Given that $e$ and $E_{t+1}$ depend on the state prediction, their errors are bounded with high probability: if $e$ is Lipschitz with constant $L_e$, then $\|e(s_{t+1}) - e(\hat{s}_{t+1})\| \leq L_e \Gamma_t$; similarly, the error in $E_{t+1}$ is bounded by $\Gamma_E \leq L_E \Gamma_t$. For simplicity, we take a uniform bound $\Gamma_t$ when $L_e, L_E \leq 1$. The set of safe actions is defined as: $\hat{A}_{\text{safe}}(s_t) = \{a \in A \mid \nu(e(\hat{f}(s_t, a)), \hat{E}_{t+1}) > 2L_\nu \Gamma_t\}$. If an action is selected from $\hat{A}_{\text{safe}}$, we guarantee $\nu(e(s_{t+1}), E_{t+1}) > 0$, ensuring a safe state at $t+1$. The final bound depends on the failure probability of state prediction and the probability of selecting safe actions. $\square$

## 5 EXPERIMENTS

We empirically evaluate our approach to assess its safety, generalization, and efficiency across diverse RL tasks. We compare against established safe RL baselines and analyze three variants of our method: using only the safety-regularized objective, only the adaptive shield, and their combination. Our experiments are guided by the following research questions:

- **RQ1:** How does our approach balance safety and task performance during training without being informed of changing hidden parameters?
- **RQ2:** How well does our approach generalize to out-of-distribution test environments by inferring varying hidden parameters online?

### 5.1 EXPERIMENTAL SETUP

**Environments.** We conduct experiments using the Safe-Gym benchmark (Ji et al., 2023) for safe RL, with two robot types: *Point* and *Car*. Each robot performs four tasks: (1) *Goal*: navigate to a target while avoiding obstacles; (2) *Button*: activate a button while avoiding hazards; (3) *Push*: push an object to a goal under contact constraints; (4) *Circle*: follow a circular path while staying within safe boundaries. Robot-task combinations are denoted as robot-task (e.g., Point-Goal, Car-Circle). Each task includes a safety constraint (e.g., obstacle avoidance or region adherence). Episode-level randomness is introduced by sampling gravity, and four hidden dynamics parameters: damping, mass, inertia, and friction.

**Baselines.** We compare our approach to six established safe RL algorithms:

- **Saute**: A state augmentation technique with safety budgets for almost sure constraint satisfaction, applicable to a wide range of RL algorithms such as PPO or RCPO (Li et al., 2022).
- **PPO-Lag**: Proximal Policy Optimization with Lagrangian updates for both reward and constraint (Schulman et al., 2017; Ray et al., 2019).
- **RCPO**: Reward Constrained Policy Optimization, which uses policy gradients to optimize a reward function penalized by safety violations (Tessler et al., 2018).
- **CPO**: Constrained Policy Optimization with joint second-order updates to enforce linearized cost constraints (Achiam et al., 2017).
- **CUP**: Constrained Update Projection, a policy optimization method that projects updates to satisfy safety constraints with theoretical guarantees (Yang et al., 2022).
- **USL**: Unrolling Safety Layer, which re-weights the policy loss for safety and projects unsafe actions into a feasible set at execution (Zhang et al., 2023).

*Baselines directly access hidden parameters $\phi$ (i.e., $b_\phi = \phi$) for dynamics adaptation*, as they do not perform inference. In contrast, our approach uses $\hat{f}_{\text{FE}}$ to infer hidden parameters online and is evaluated *without this privileged information*, demonstrating robustness under limited parameter awareness. We use RCPO without access to the hidden parameters $\phi$ as the base RL algorithm. On top of this, we evaluate RCPO combined with SRO, RCPO combined with Shield, and RCPO combined with both SRO and Shield. For brevity, we refer to these as SRO, Shield, and SRO + Shield, respectively. We also provide results using PPO-Lag combined with our methods in Appendix G.

**Hyperparameters.** All methods use the default hyperparameters provided by their respective implementations: Omni-Safe (Ji et al., 2024) for Saute, PPO-Lag, RCPO, CPO, and CUP, and Safe-

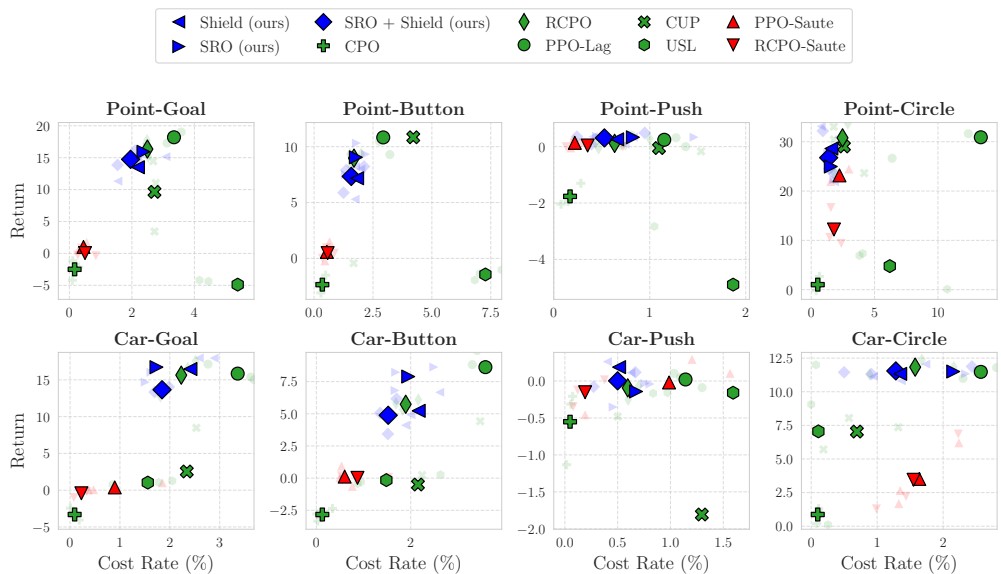

Figure 1: Results display the mean reward and cost rate (%) over the last 20 epochs across seeds. The top-left position is desirable, indicating higher returns with lower cost rates. Solid points represent mean return and cost rate, while transparent points depict individual seed results.

RL-Kit (Zhang et al., 2023) for USL. When evaluating our approach on top of each base algorithm (e.g., RCPO), we adopt the same hyperparameters as the corresponding baseline to ensure a fair comparison. Each method is trained for 2 million environment steps using 3 random seeds. Each trained policy is evaluated over 100 episodes at test time.

We set the pre-safety distance to 0.275 and the ACP failure probability to 2%. The function encoder $\hat{f}_{\text{FE}}$ is pre-trained on 1000 episodes (1000 steps each) collected by a trained PPO policy. The $\hat{f}_{\text{FE}}$ remains fixed during policy training, introducing realistic prediction error that is managed by ACP. All agents are trained under a strict safety constraint, with a cost limit of zero.

For training, environment parameters $\phi$ (gravity, damping, mass, inertia, friction) are sampled uniformly from the interval $[0.3, 1.7]$. For out-of-distribution evaluation, the parameters are sampled from the interval $[0.15, 0.3] \cup [1.7, 2.5]$, and the number of obstacles is increased to stress generalization.

**Metrics.** We evaluate each method using per-episode averages for the following metrics, each capturing a different aspect of performance: (1) *Return*, measuring task performance as the cumulative reward per episode; (2) *Cost Rate*, reflecting safety by measuring the frequency of constraint violations per timestep.

## 5.2 RESULTS ANALYSIS

**RQ1: Trade-offs Between Safety and Return.** Figure 1 shows the episodic return and cost rate across four tasks during training. Baseline methods exhibit a range of trade-offs. PPO-Lag tends to achieve high returns but incur higher cost rates. CPO, PPO-Saute, and RCPO-Saute enforce strict safety via a zero-violation constraint, often sacrificing reward learning, which leads to suboptimal policies in multiple tasks. USL, dependent on cost-Q-value estimation, underperforms across all tasks due to its sensitivity to environmental stochasticity, such as randomly reset obstacle positions and dynamic changes, which disrupt cost estimation. CUP preserves task-solving performance but frequently violates strict cost limits. RCPO maintains a reasonable balance between safety and return but struggles to meet lower cost thresholds. These results highlights the challenge of balancing safety and returns, even when hidden parameters are provided as inputs, in the presence of varying hidden parameters. To assess the impact of providing fixed parameters, we compare performance with and without fixed parameters. See Appendix L for details.

In contrast, our methods (using RCPO as the base RL algorithm) consistently achieve lower cost rates while maintaining competitive returns, demonstrating their ability to balance safety and task

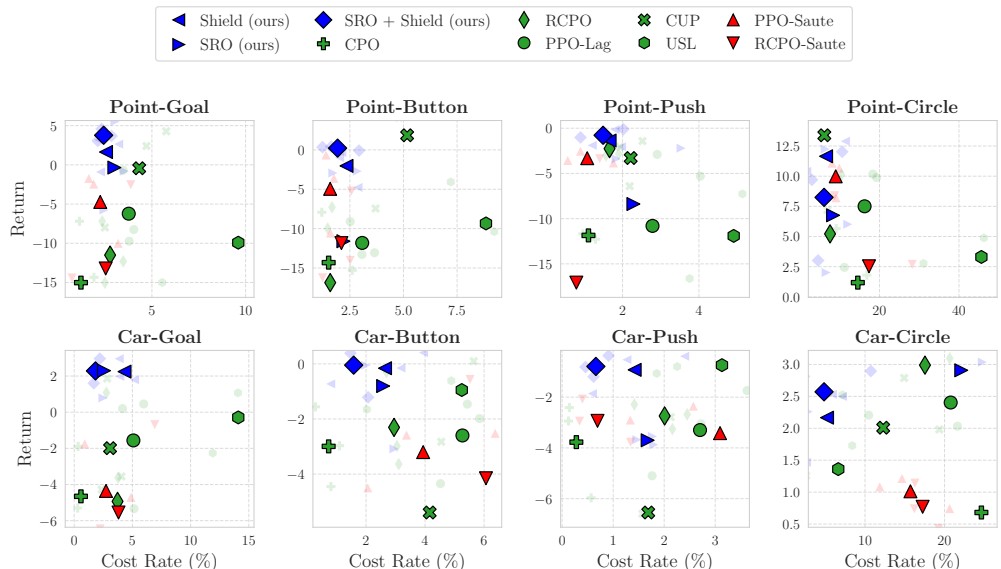

Figure 2: Trade-off between average episodic return and cost rate in out-of-distribution domains. The top-left position is desirable, indicating higher returns with lower cost rates. Solid points represent mean return and cost rate, while transparent points depict individual seed results.

performance during training. Variants using only safety-regularized objective (SRO) or only adaptive shield also reduce cost violations compared to baselines, but are less effective than the combined method. We observe similar results when using PPO as the base RL algorithm for our methods (see results in G).

*Takeaway: Our combined method (SRO + shield) achieves an effective reward–cost trade-off during training and remains consistently robust to unseen variations in environmental parameters.*

**RQ2: Generalization to Out-of-Distribution Environments.** Figure 2 illustrates the trade-off between average episodic return and cost rate in out-of-distribution test environments across all tasks. Each marker corresponds to the mean performance across three trained policies (one per seed) for a given method, evaluated separately on Point and Car robots.

Our full method (SRO + Shield) consistently appears near the desirable position (high return and low cost rate) across all tasks, indicating strong generalization to previously unseen dynamics. SRO-only and Shield-only variants also perform well but tend to deviate more from the desirable position. Shield typically remains stable because its logic, which filters unsafe actions based on predicted states and safety measures like proximity to obstacles, holds regardless of OOD conditions. This trend highlights the complementary effect of combining proactive (SRO) and reactive (Shield) safety mechanisms.

Among the baselines, CPO and Saute maintain low cost rates but sacrifice return, often positioning them outside the desirable region. The remaining algorithms exhibit inconsistent performance across environments, lacking consistent patterns.

*Takeaway: our approach generalizes effectively to out-of-distribution settings, consistently achieving a favorable balance between return and safety across robot types and task variations.*

**Ablation Studies.** We conduct additional ablation studies, presented in the appendix. These include: comparing runtime overhead for executing shielding (Appendix C); Analyzing the connection between our shielding mechanism and control theory (Appendix E); Analyzing key hyperparameters like sampling size and safety bonus (Appendix F); Applying our method to other base RL algorithms, such as PPO-Lag (Appendix G); Evaluating the impact of the function encoder representation on both task performance and dynamics prediction accuracy compared to alternative predictors (Appendices H and I); and providing additional experiments on HalfCheetah (Appendix J). Our ablations show that augmenting RL algorithms with shielding or SRO enhances safety without compromising performance, which remains stable across varying sampling sizes and safety bonus

values. Additionally, our function encoder-based inference effectively adapts to changing transition dynamics $T_\phi$, achieving performance comparable to oracle representations.

## 6 CONCLUSION

We presented a novel approach for safe and generalizable reinforcement learning in settings with dynamically varying hidden parameters. Our approach comprises three key components: (1) a safety-regularized objective that promotes low-violation behavior during training, (2) function encoder-based inference of hidden dynamics, and (3) an adaptive runtime shield that uses conformal prediction to filter unsafe actions based on uncertainty at execution time. Experimental results demonstrate that our approach consistently outperforms baselines in reducing safety violations while maintaining competitive task performance, and generalizes effectively across diverse tasks and out-of-distribution environments.

Despite its effectiveness, our approach has several limitations. First, the safety guarantees rely on assumptions about the structure of the cost function, although these apply to a broad range of practical scenarios. Second, the method depends on an offline dataset to train the function encoder, which may limit applicability in settings without prior data. Third, our evaluation has so far been limited to simulated environments. Future work will aim to address these limitations by relaxing modeling assumptions, reducing reliance on offline data, and extending evaluations to physical robotic platforms to assess scalability and real-world applicability.

## ETHICS STATEMENT

This research adheres to the ICLR Code of Ethics (https://iclr.cc/public/CodeOfEthics). Our reinforcement learning framework, incorporating runtime shielding and safety-regularized objectives, was evaluated in simulated Safe-Gym environments, ensuring no involvement of human subjects, sensitive data, or real-world deployment that could pose ethical risks. We prioritized safety and robustness by addressing generalization to unseen dynamics, with experiments designed to minimize potential unsafe behavior. No conflicts of interest or external funding influenced this work. A large language model was used solely to polish text and assist in generating visualizations, with all core research and results developed independently by the authors. We uphold the highest standards of research integrity and transparency as outlined in the ICLR guidelines.

## REPRODUCIBILITY STATEMENT

To ensure reproducibility of our results, we provide the complete source code, including all hyperparameters, training scripts, evaluation protocols, and shielding mechanisms, via an anonymous repository: https://osf.io/pc2fg/files/osfstorage?view_only=7d9c265765074d59a6cdecdbce6b66aa. All theoretical results, including proofs of probabilistic safety guarantees, are included in Appendix A, with clear explanations of assumptions and derivations. These materials collectively ensure that our findings can be independently verified and reproduced.

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

# A  PROOFS

This section presents our main theoretical results, including proofs. We first introduce the necessary notations.

**Notations.** We introduce the notation for dimensions and sets as follows. For $n, m, k_1, k_2 \in \mathbb{N}$, let $n$ denote the state dimension, $m$ the action dimension, $k_1$ the hidden parameter dimension, and $k_2$ the dimension of the function encoder's learned representation. Note that the dimension $k_2$ of the learned representation may differ from $k_1$, depending on the number of chosen basis functions. The state space is $S \subseteq \mathbb{R}^n$, the action space is $A \subseteq \mathbb{R}^m$, the hidden parameter space is $\Phi \subseteq \mathbb{R}^{k_1}$, and the learned representation space is $\mathcal{B} \subseteq \mathbb{R}^{k_2}$, where $\mathcal{B}$ is the coefficient space of basis functions for the function encoder.

- $\mathcal{M}$: Constrained Hidden Parameter Markov Decision Process.
- $s_t \in \mathbb{R}^n$: State at time step $t$.

- $\hat{s}_t \in \mathbb{R}^n$: Predicted State for time step $t$.

- $X_{i,t}\ (X_t) \in \mathbb{R}^2$: Position of the $i$-th obstacle at time step $t$. The index $i$ is omitted when referring to a single obstacle without ambiguity.

- $\hat{X}_{i,t}\ (\hat{X}_t) \in \mathbb{R}^2$: Predicted position of the $i$-th obstacle at time step $t$. The index $i$ is omitted when referring to a single obstacle without ambiguity.

- $a_t \in \mathbb{R}^m$: Action at time step $t$.

- $\hat{f} : S \times A \to S$: Continuous transition dynamics predictor.

- $\phi \in \mathbb{R}^{k_1}$: hidden parameter.

- $b_\phi \in \mathbb{R}^{k_2}$: learned representation to $T_\phi$.

- $s' \sim T(\cdot \mid s, a, \phi)$: Transition dynamics given parameter $\phi$. When $s, a, s'$ are unspecified, we denote this by $T_\phi$.

- $C : S \times A \times S \to [0, 1]$: Cost function bounded in $[0, 1]$.

- $\mathcal{S}_{\text{safe}}(s_t, a_t)$: Safe state set, defined as $\{s_t \in \mathcal{S} \mid C(s_t, a_t, s_{t+1}) = 0 \text{ s.t } T(s_{t+1} \mid s_t, a_t, \phi) > 0\}$ where $\phi$ is a parameter sampled per episode.

- $Q_R^\pi(s, a, b_\phi)$: State-action value function for reward under policy $\pi$ defined as

$$\mathbb{E}_{\pi, T_\phi} \left[ \sum_{t=0}^{\infty} \gamma^t R(s_t, a_t, s_{t+1}) \mid s_0 = s, a_0 = a, \phi \right].$$

We aim to maximize this value.

- $Q_C^\pi(s, a, b_\phi)$: State-action value function for cost under policy $\pi$ defined as

$$\mathbb{E}_{\pi, T_\phi} \left[ \sum_{t=0}^{\infty} \gamma^t C(s_t, a_t, s_{t+1}) \mid s_0 = s, a_0 = a, \phi \right].$$

We aim to minimize this value.

- $Q_{\text{aug}}^\pi(s, a, b_\phi)$: Safety-regularized state-action value function defined as $Q_R^\pi(s, a, b_\phi) + \alpha Q_{\text{safe}}^\pi(s, a, b_\phi)$ where $\alpha$ is a positive constant and

$$Q_{\text{safe}}^\pi(s, a, b_\phi) = -\frac{\int_{B(a,\epsilon) \cap A} \pi(x \mid s, b_\phi) Q_C^\pi(s, x, b_\phi) dx}{V_C^\pi(s, b_\phi) + \epsilon}$$

where $\epsilon > 0$ is a small constant ensuring numerical stability, and $B(a, \epsilon)$ denotes a small ball of radius $\epsilon$ centered at $a$. We aim to maximize this value, whose maximum value is 0.

- $J_{\text{safe}}(\pi)$: Safety-regularizer for policy $\pi$ defined as

$$\mathbb{E}_{\phi \sim P_\phi, s_0 \sim \mu_0(\cdot \mid \phi), a_0 \sim \pi(\cdot \mid s_0, b_\phi)} \left[ Q_{\text{safe}}^\pi(s_0, a_0, b_\phi) \right].$$

The policy $\pi$ aims to maximize this value.

- $J_{\text{aug}}(\pi)$: Safety-regularized objective function for policy $\pi$ defined as

$$\mathbb{E}_{\phi \sim P_\phi, s_0 \sim \mu_0(\cdot \mid \phi), a_0 \sim \pi(\cdot \mid s_0, b_\phi)} \left[ Q_{\text{aug}}^\pi(s_0, a_0, b_\phi) \right].$$

The policy $\pi$ aims to maximize this value.

- $J_R(\pi)$: Standard reward objective function for policy $\pi$ defined as

$$\mathbb{E}_{\phi \sim P_\phi, s_0 \sim \mu_0(\cdot \mid \phi), a_0 \sim \pi(\cdot \mid s_0, b_\phi)} \left[ Q_R^\pi(s_0, a_0, b_\phi) \right].$$

The policy $\pi$ aims to maximize this value.

- $J_C(\pi)$: Standard cost objective function for policy $\pi$ defined as

$$\mathbb{E}_{\phi \sim P_\phi, s_0 \sim \mu_0(\cdot \mid \phi), a_0 \sim \pi(\cdot \mid s_0, b_\phi)} \left[ Q_C^\pi(s_0, a_0, b_\phi) \right].$$

The policy $\pi$ aims to minimize this value.

- $\Pi_{\text{zero-violation}}$: Set of zero-violation policies defined as $\{\pi \mid J_C(\pi) = 0\}$.

- $\delta \in (0, 1)$: Failure probability.

- $\Gamma_t \in \mathbb{R}^+$: Adaptive Conformal Prediction (ACP) threshold at time step $t$.
- $\xi^\pi(s, \phi)$: Average cost under policy $\pi$ defined as

$$\xi^{\pi^*}(s, \phi) = \lim_{H \to \infty} \frac{1}{H} \mathbb{E}_{\pi^*, T_\phi} \left[ \sum_{t=0}^{H-1} C(s_t, a_t, s_{t+1}) \mid s_0 = s, \phi \right],$$

starting from state $s$ and parameter $\phi$.

- $\pi^*$: Optimal policy satisfying constraint on the average cost rate.

We restate our proposition and theorems, then provide detailed proofs.

**Proposition 1.** Let $\Pi_{\text{zero-violation}}$ be the set of zero-violation policies. Then, for any $\alpha \geq 0$, the optimal policy obtained by maximizing the safety-regularized objective function $J_{\text{aug}}(\pi)$ within $\Pi_{\text{zero-violation}}$ is equivalent to the optimal policy obtained by maximizing the standard reward objective $J_R(\pi)$ within the same set of policies.

*Proof.* By definition, if $\pi \in \Pi_{\text{zero-violation}}$, then for any state $s$, parameter $\phi$, and action $a$ with $\pi(a|s, b_\phi) > 0$, the state-action value function for the cost is zero: $Q_C^\pi(s, a, b_\phi) = 0$. Hence, any action sampled around $a$ by the policy $\pi$ will have $Q_C^\pi(s, a, b_\phi) = 0$ leading to

$$\int_{B(a, \epsilon) \cap A} \pi(x \mid s, b_\phi) Q_C^\pi(s, x, b_\phi) dx = 0.$$

This implies that for such policies, the safety term $Q_{\text{safe}}^\pi(s, a, b_\phi)$ in the regularized objective is a constant value of $0$.

Therefore, for any policy $\pi \in \Pi_{\text{zero-violation}}$, the safety-regularized objective function becomes:

$$\begin{aligned} J_{\text{aug}}(\pi) &= \mathbb{E}_{\phi \sim P_\phi, s_0 \sim \mu_0(\cdot|\phi), a_0 \sim \pi(\cdot|s_0, b_\phi)} \left[ Q_R^\pi(s_0, a_0, b_\phi) + \alpha \cdot Q_{\text{safe}}^\pi(s_0, a_0, b_\phi) \right] \\ &= \mathbb{E}_{\phi \sim P_\phi, s_0 \sim \mu_0(\cdot|\phi), a_0 \sim \pi(\cdot|s_0, b_\phi)} \left[ Q_R^\pi(s_0, a_0, b_\phi) + \alpha \cdot 0 \right] \\ &= J_R(\pi) \end{aligned}$$

Thus, maximizing $J_{\text{aug}}(\pi)$ within $\Pi_{\text{zero-violation}}$ is equivalent to maximizing $J_R(\pi)$ within the same set. $\square$

Next, we will prove our main theorem. To prove the main theorem, we first establish the necessary notation and settings.

We define the safe state set for a given state $s \in S$, action $a \in A$, and hidden parameter $\phi \in \Phi$ as: $\mathcal{S}_{\text{safe}}(s, a) = \{s' \in S \mid C(s, a, s') = 0 \text{ s.t. } T(s' \mid s, a, \phi) > 0\}$. Thus, $\mathcal{S}_{\text{safe}}(s, a)$ contains next states $s'$ where the safety condition is satisfied. Throughout the theorem, we address a cost function defined by safe distance

$$C(s_t, a_t, s_{t+1}) = \begin{cases} 1 & \text{if } \min_{X_{t+1}} \|pos(s_{t+1}) - X_{t+1}\| \leq d \\ 0 & \text{otherwise,} \end{cases}$$

where $X_{t+1}$ denotes the positions of obstacles at time step $t+1$, and $pos(s_{t+1})$ represents the agent's position in state $s_{t+1}$. As our theorem applies to any norm satisfying the triangle inequality, we do not specify a particular norm.

We assume the state includes the agent's position, a natural choice for navigation tasks. Generally, the state contains critical information needed to evaluate the cost function. Hence, a function $pos : S \to \mathbb{R}^2$ is a projection mapping, which is 1-Lipschitz continuous, meaning that

$$\|pos(s) - pos(s')\| \leq \|s - s'\|$$

for all $s, s' \in S$.

We assume that the obstacle position $X_t$ can be derived from the state $s_t$. This assumption is reasonable, as the agent's state typically includes safety-critical information. For instance, robots in navigation tasks use sensors to detect nearby obstacles, with this information integrated into the agent's state. This setup applies to all navigation environments in Safety Gymnasium. Formally, we assume a

1-Lipschitz continuous function sensor : $S \to \mathbb{R}^{2M}$, defined as $\text{sensor}(s_t) = [X_{1,t}, X_{2,t}, \ldots, X_{M,t}]$, where $X_{i,t} \in \mathbb{R}^2$ is the position of the $i$-th obstacle detected by the robot, and $M$ is the number of detected obstacles. When more than $M$ obstacles are present, the sensor typically detects the $M$ closest ones.

We adopt a Gaussian policy $\pi$, commonly employed in training RL policies across various algorithms.

**Remark 1** (Exchangeability of Episode Data). The validity of conformal prediction depends on the assumption that the calibration and test data are exchangeable. In our sequential decision-making context, this requires careful consideration. We ensure this property by treating each episode as an independent data-generating process governed by a transition dynamics model $T_\phi$ given a specified parameter $\phi$. Specifically, a hidden parameter $\phi$ is sampled at the beginning of each episode, and this hidden parameter remains fixed for the episode's entire duration. The dataset collected within this episode is a sequence of transition tuples, $D_{\text{ep}} = \{(s_0, a_0, s_{t+1}), (s_1, a_1, s_2), \ldots\}$. While the sequence of states is temporally dependent, the individual transition tuples are conditionally independent and identically distributed (i.i.d.) given the parameter $\phi$. That is, each next state $s_{t+1}$ is drawn independently from the distribution $T_\phi(\cdot \mid s_t, a_t)$, a process that is identical for all steps $t$ within the episode. Formally, the joint probability of observing the sequence of transitions in $D_{ep}$ conditioned on $\phi$ is given by $\mathbb{P}(D_{ep} \mid \phi) = \mu(s_0 \mid \phi) \prod_{t=0}^{T-1} T_\phi(s_{t+1} \mid s_t, a_t)$. Because the product operator is commutative, the joint probability is invariant to any permutation of the transition tuples in the sequence. This conditional i.i.d. property implies that the sequence of transition tuples is exchangeable. Our online calibration procedure adheres to this principle. By collecting the calibration set from the initial steps of the same episode, we guarantee that both the calibration data and the subsequent test data (within that episode) are drawn from the same distribution $T_\phi$ given $\phi$, thereby satisfying the exchangeability assumption required for valid conformal prediction. Thereby, during the first 100 steps, we gather samples for calibration without using ACP region. After 100 steps, we employ the online-collected calibration set to determine ACP region.

Our argument extends to any Lipschitz continuous cost function bounded in $[0, 1]$, with the proof following a similar approach. If the cost function is bounded by a constant $D > 1$, the proof remains valid, but the final bound is scaled by $D$.

**Lemma 1.** Let $\hat{f}$ be a transition dynamics predictor and $e(a) = \min_{\hat{X}_{t+1}} \|\text{pos}(\hat{f}(s_t, a)) - \hat{X}_{t+1}\|$. Under the adaptive shielding mechanism with sampling size $N$ for each episode with parameter $\phi$, one of the following conditions holds:

1. $\mathbb{P}(s_{t+1} \in S_{\text{safe}}(s_t, a_t)) \geq 1 - \delta$, where $s_{t+1} \sim T(\cdot \mid s_t, a_t, \phi)$,

2. $\min_{X_{t+1}} \|\text{pos}(s_{t+1}) - X_{t+1}\| \geq \max_{a \in A} e(a) - \epsilon_N - 2\Gamma_t$,

where $\lim_{N \to \infty} \epsilon_N = 0$ and $\Gamma_t$ is the ACP confidence region for the state prediction at time step $t + 1$.

*Proof.* Note that $s_{t+1}$ is safe if $\min_{X_{t+1}} \|\text{pos}(s_{t+1}) - X_{t+1}\| > d$. The ACP gives us a probabilistic bound on the deviation between the true next state $s_{t+1}$ and the predicted state $\hat{s}_{t+1} = \hat{f}(s_t, a_t)$ :

$$\mathbb{P}(\|\hat{s}_{t+1} - s_{t+1}\| \leq \Gamma_t) \geq 1 - \delta$$

We connect the safety of $s_{t+1}$ to the position of the predicted state $\hat{s}_{t+1}$, using this bound. By triangle inequality, we have

$$\min_{X_{t+1}} \|\text{pos}(s_{t+1}) - X_{t+1}\| \geq \min_{X_{t+1}} \|\text{pos}(\hat{s}_{t+1}) - \hat{X}_{t+1}\| - \|\hat{X}_{t+1} - X_{t+1}\| - \|\text{pos}(s_{t+1}) - \text{pos}(\hat{s}_{t+1})\|.$$

$$(13)$$

Since pos function and sensor function are 1-Lipschitz, we have

$$\|\text{pos}(s_{t+1}) - \text{pos}(\hat{s}_{t+1})\| \leq \|s_{t+1} - \hat{s}_{t+1}\| \quad \text{and} \quad \|X_{t+1} - \hat{X}_{t+1}\| \leq \|s_{t+1} - \hat{s}_{t+1}\|.$$

Hence, if $\|s_{t+1} - \hat{s}_{t+1}\| \leq \Gamma_t$ (which occurs with probability at least $1 - \delta$), then

$$\|\text{pos}(s_{t+1}) - \text{pos}(\hat{s}_{t+1})\| \leq \Gamma_t \text{ and } \|X_{t+1} - \hat{X}_{t+1}\| \leq \Gamma_t.$$

This implies

$$\min_{X_{t+1}} \|\text{pos}(s_{t+1}) - X_{t+1}\| \geq \min_{X_{t+1}} \|\text{pos}(\hat{s}_{t+1}) - X_{t+1}\| - 2\Gamma_t. \tag{14}$$

Thus, if

$$h(a_t) = \min_{X_{t+1}} \left\| \text{pos}(\hat{s}_{t+1}) - \hat{X}_{t+1} \right\| > d + 2\Gamma_t,$$

then $\min_{X_{t+1}} \|\text{pos}(s_{t+1}) - X_{t+1}\| > d$ whenever $\|s_{t+1} - \hat{s}_{t+1}\| \leq \Gamma_t$. Let us define the set of safe actions on the predicted state by

$$\hat{A}_{\text{safe}}(s_t) = \{a \in A \mid e(a) > d + 2\Gamma_t\}.$$

We now consider two cases based on the feasibility of selecting an action from the set $\hat{A}_{\text{safe}}$.

**Case 1:** If we can select $a_t \in \hat{A}_{\text{safe}}(s_t)$ and $\|s_{t+1} - \hat{s}_{t+1}\| \leq \Gamma_t$, then $s_{t+1}$ is safe by Equation 14. By ACP of our adaptive shielding mechanism, we guarantee

$$\mathbb{P}\left(\|\hat{s}_{t+1} - s_{t+1}\| \leq \Gamma_t\right) \geq 1 - \delta$$

where $\delta$ is a failure probability of ACP. Thus, condition 1 holds.

**Case 2:** If we cannot select $a_t \in \hat{A}_{\text{safe}}(s_t)$, our adaptive shielding mechanism samples $N$ actions $\{a_t^{(i)}\}$ and picks the action $a_t$ such $e(a_t) = \max_{a \in \{a_t^{(i)}\}} e(a)$. Note that $e(a)$ is continuous on $a$ and a Gaussian policy $\pi$ assigns positive probability to any subset of action space $A$. Hence, as sample size $N$ goes to $\infty$, $\max_{a \in A} e(a) - e(a_t) = \epsilon_N$ goes to 0. Also, by Equation 14, we have

$$\|\text{pos}(s_{t+1}) - X_{t+1}\| \geq e(a_t) - 2\Gamma_t = \max_{a \in A} e(a) - \epsilon_N - 2\Gamma_t.$$

Thus, condition 2 holds. $\qquad\square$

To prove the theorem, we recall the function $e(a) = \min_{\hat{X}_{t+1}} \|\text{pos}(\hat{f}(s_t, a)) - \hat{X}_{t+1}\|$, representing the minimum distance between the predicted state and predicted obstacles. Using this, we define the safe action set for the predicted state as:

$$\hat{A}_{\text{safe}}(s_t) = \{a \in A \mid e(a) > d + 2\Gamma_t\}.$$

Lemma 1 considers two cases based on whether sampling from $\hat{A}_{\text{safe}}(s_t)$ is feasible. To derive a bound for the average cost rate constraint, we analyze both cases by defining $\epsilon_t = \mathbb{P}(\hat{A}_{\text{safe}}(s_t) = \emptyset)$. We assume that if $\hat{A}_{\text{safe}}(s_t)$ is non-empty, a large sample size $N$ allows sampling an action from this set, as discussed in Lemma 1.

**Theorem 1.** Given a Constrained Hidden Parameter MDP $\mathcal{M} = (S, A, \Phi, T, R, C, \gamma, P_\phi)$ with initial state $s_0 \in S$, and failure probability $\delta \in (0, 1)$, an optimal policy $\pi^* : S \times \Phi \to A$, augmented with an adaptive shield, maximizes the expected cumulative discounted return $J_R(\pi^*)$, while satisfying the average cost rate constraint:

$$\xi^{\pi^*}(s, \phi) = \lim_{H \to \infty} \frac{1}{H} \mathbb{E}_{\pi^*, T_\phi} \left[ \sum_{t=0}^{H-1} C(s_t, a_t, s_{t+1}) \mid s_0 = s, \phi \right] \leq \delta + \bar{\epsilon}(1 - \delta), \tag{15}$$

for some $0 \leq \bar{\epsilon} \leq 1$ and $\phi \sim P_\Phi$.

*Proof.* At each time step $t$, $\hat{A}_{\text{safe}}(s_t)$ is non-empty with probability $1 - \epsilon_t$, allowing us to sample actions with a large sample size $N$. By Lemma 1, this guarantees:

$$\mathbb{P}\left(s_{t+1} \in S_{\text{safe}}(s_t, a_t)\right) \geq 1 - \delta$$

where $s_{t+1} \sim T(\cdot \mid s_t, a_t, \phi)$, and the safe state set is defined as:

$$S_{\text{safe}}(s_t, a_t) = \{s' \in S \mid C(s_t, a_t, s') = 0, T(s' \mid s_t, a_t, \phi) > 0\}$$

Thus, when $\hat{A}_{\text{safe}}(s_t)$ is non-empty with probability $1 - \epsilon_t$, the cost function satisfies:

$$C\left(s_t, a_t, s_{t+1}\right) = \begin{cases} 1 & \text{with probability at most } \delta \\ 0 & \text{with probability at least } 1 - \delta \end{cases},$$

which implies $\mathbb{P}(C = 1) \leq \delta$ and $\mathbb{P}(C = 0) \geq 1 - \delta$. In this case, the expected cost per step is bounded as follows:

$$\mathbb{E}_{\pi^*}\left[C\left(s_t, a_t, s_{t+1}\right)\right] \leq \delta(1 - \epsilon_t).$$

When $\hat{A}_{\text{safe}}(s_t)$ is empty with probability $\epsilon_t$, the expected cost per step is bounded as follows:

$$\mathbb{E}_{\pi^*}\left[C\left(s_t, a_t, s_{t+1}\right)\right] \leq \epsilon_t.$$

Combining both cases, the expected cost per step is bounded by:

$$\mathbb{E}_{\pi^*}\left[C\left(s_t, a_t, s_{t+1}\right)\right] \leq \delta(1 - \epsilon_t) + \epsilon_t = \delta + \epsilon_t(1 - \delta).$$

By the linearity of expectation, this per-step bound extends to the long-term average cost for a fixed parameter $\phi$:

$$\xi^{\pi^*}\left(s_0, \phi\right) = \lim_{H \to \infty} \frac{1}{H} \sum_{t=0}^{H-1} \mathbb{E}_{\pi^*, T_\phi}\left[C\left(s_t, a_t, s_{t+1}\right)\right] \tag{16}$$

$$\leq \limsup_{H \to \infty} \frac{1}{H} \sum_{t=0}^{H-1} \left(\delta + \epsilon_t(1 - \delta)\right) \tag{17}$$

$$= \delta + \bar{\epsilon}(1 - \delta) \tag{18}$$

where $\bar{\epsilon} = \limsup_{H \to \infty} \frac{1}{H} \sum_{t=0}^{H-1} \mathbb{E}\left[\epsilon_t\right]$. This satisfies Equation 15, completing the proof. Moreover, if safe actions exist at each time step $t$, i.e., $\epsilon_t = \mathbb{P}(\hat{A}_{\text{safe}} = \emptyset) = 0$, $\bar{\epsilon}$ becomes 0. Hence, we can bound the equation with a small failure probability $\delta$. $\qquad\square$

# B  THEORETICAL GUARANTEES FOR SAFETY-REGULARIZED TRPO AND CPO

This section provides a formal extension of the monotonic improvement guarantee of Trust Region Policy Optimization (TRPO) (Schulman et al., 2015) to our proposed safety-regularized objective, $J_{\text{aug}}(\pi)$. We first recap the foundational theorem of TRPO and then prove that this guarantee directly applies to our augmented objective. Finally, we provide a rigorous analysis of the trade-off between reward and safety that this guarantee implies

We now prove that this same guarantee holds for our safety-regularized objective, $J_{\text{aug}}(\pi)$. The core insight is that the TRPO proof structure is agnostic to the definition of the reward function; it depends only on the MDP dynamics and the relationship between the policies. Our method can be viewed as replacing the standard reward with an augmented reward signal.

We formally define the augmented objective as $J_R(\pi) = \mathbb{E}\left[Q_R^\pi\right]$, $J_{\text{safe}}(\pi) = \mathbb{E}[Q_{\text{safe}}^\pi]$ and $J_{\text{aug}}(\pi) = \mathbb{E}[Q_{\text{aug}}^\pi]$. We restate the policy improvement guarantee of TRPO with SRO. During the optimization, KL constraint plays the same role as in standard TRPO. This only limits how far the updated policy is allowed to move from the current policy in one step, and is independent of whether we optimize the original reward $J_R$ or our safety-augmented objective $J_{\text{aug}}$.

**Theorem 2 (Schulman et al., 2015).** Let $L_\pi^{\text{aug}}\left(\tilde{\pi}\right) = J_{\text{aug}}(\pi) + \mathbb{E}_{\phi \sim P_\phi s \sim \rho_\pi, a \sim \tilde{\pi}}\left[A_{\text{aug}}^\pi\left(s, a, b_\phi\right)\right]$. The performance of the new policy $\tilde{\pi}$ is lower-bounded by:

$$J_{\text{aug}}\left(\tilde{\pi}\right) \geq L_{\text{aug}}^\pi(\tilde{\pi}) - C_{\text{aug}} \cdot D_{KL}^{\max}(\pi, \tilde{\pi})$$

where $C_{\text{aug}} = \frac{2\gamma}{(1-\gamma)^2} \max_{s,a,\phi} \left|A_{\text{aug}}^\pi\left(s, a, b_\phi\right)\right|$.

**Monotonic improvement condition..** If the update $\tilde{\pi}$ satisfies

$$L_\pi^{\text{aug}}(\tilde{\pi}) - C_{\text{aug}} D_{KL}^{\max}(\pi, \tilde{\pi}) \geq J_{\text{aug}}\left(\pi\right),$$

then $J_{\text{aug}}\left(\tilde{\pi}\right) \geq J_{\text{aug}}\left(\pi\right)$. In other words, any update that sufficiently increases the surrogate while keeping KL small yields non-decreasing augmented performance.

**Analysis of the Reward-Safety Trade-off.** Let $\Delta J_R = J_R(\tilde{\pi}) - J_R(\pi)$ and $\Delta J_{\text{safe}} = J_{\text{safe}}(\tilde{\pi}) - J_{\text{safe}}(\pi)$. Under the monotonic-improvement condition above, for any valid policy update step:

$$\Delta J_R + \alpha \cdot \Delta J_{\text{safe}} \geq 0$$

This inequality provides a formal characterization of the trade-off between reward and safety.

**Bounded Reward Degradation for Safety Improvement.** If an update improves the safety objective ($\Delta J_{\text{safe}} > 0$), we can rearrange the inequality to bound the permissible change in reward:

$$\frac{\Delta J_R}{\Delta J_{\text{safe}}} \geq -\alpha \quad \implies \quad \Delta J_R \geq -\alpha \cdot \Delta J_{\text{safe}}$$

This proves that for a given gain in safety, the reward is guaranteed not to decrease by more than $\alpha$ times that gain. The hyperparameter $\alpha$ thus acts as a maximum acceptable cost in reward for a unit of safety improvement.

**Bounded Safety Degradation for Reward Improvement.** Conversely, if an update improves the reward objective ($\Delta J_R > 0$), we can bound the permissible change in the safety term:

$$\Delta J_{\text{safe}} \geq -\frac{1}{\alpha} \Delta J_R$$

This proves that for a given gain in reward, the safety term is guaranteed not to decrease by more than $1/\alpha$ times that gain. This demonstrates that the algorithm will forgo policy updates that yield high rewards at the expense of excessive safety violations, where the threshold for excessive is explicitly controlled by $\alpha$.

Now, we also consider safety perspectives of SRO. We mainly analyze how SRO affects safety perspective considering theorems from CPO (Achiam et al., 2017).

**Relation to CPO-style constraint guarantees.** CPO's worst-case bound (Achiam et al., 2017) depends only on KL constraint and cost advantage $A_C^{\pi_k}$. This is agnostic to how the updated policy $\pi_{k+1}$ is produced. Therefore, the bound from Proposition 2 in Achiam et al. (2017)

$$J_C\left(\pi_{k+1}\right) \leq J_C\left(\pi_k\right) + \frac{\sqrt{2\delta}\gamma}{(1-\gamma)^2} \epsilon_C^{\pi_{k+1}}$$

remains the same with $J_{\text{aug}}$. The sole role of SRO is to restrict the updated policy $\pi_{k+1}$ so that the quantity $\epsilon_C^{\pi_{k+1}} = \max_s \left| \mathbb{E}_{a \sim \pi_{k+1}}[A_C^{\pi_k}(s,a)] \right|$ remains small. Because SRO penalizes actions with high estimated long-term cost, we can upper-bound $\epsilon_C^{\pi_{k+1}}$ in terms of the safety weight $\alpha$. Substituting this bound into the CPO inequality provided a worst-case cost guarantee for the SRO-TRPO update.

Throughout this section, we assume that the ball $B(a, \epsilon)$ is small enough that local averages approximate point values, i.e., $Q_{\text{safe}}^\pi(s,a) \approx -\frac{Q_C^\pi(s,a)}{V_C^\pi(s)+\epsilon}$. We begin with introducing necessary Lemma for bound of $A_{\text{aug}}^\pi$.

**Lemma 2.** Let $A_{\text{aug}}^\pi(s,a) = A_R^\pi(s,a) + \alpha A_{\text{safe}}^\pi(s,a)$, where $A_R^\pi(s,a) = Q_R^\pi(s,a) - V_R^\pi(s)$,

$$A_{\text{safe}}^\pi(s,a) = Q_{\text{safe}}^\pi(s,a) - V_{\text{safe}}^\pi(s), \text{ and } V_{\text{safe}}^\pi(s) = \mathbb{E}_{a' \sim \pi(\cdot|s)}\left[Q_{\text{safe}}^\pi(s,a')\right].$$

Then, $Q_{\text{safe}}^\pi(s,a) \in (-1, 0]$ for all $(s,a)$, and $\left|A_{\text{aug}}^\pi(s,a)\right| \leq |A_R^\pi(s,a)| + \alpha$

*Proof.* By design, $Q_{\text{safe}}^\pi(s,a) \in (-1, 0]$. Consequently, the value function $V_{\text{safe}}^\pi(s)$, being an expectation of $Q$, is also in $(-1, 0]$. The advantage is defined as $A_{\text{safe}}^\pi(s,a) = Q_{\text{safe}}^\pi(s,a) - V_{\text{safe}}^\pi(s)$. The maximum possible value is $0 - (-1) = 1$ (when $Q = 0, V = -1$). The minimum possible value is $-1 - 0 = -1$ (when $Q = -1, V = 0$). Thus, $|A_{\text{safe}}^\pi(s,a)| \leq 1$. By triangular inequality, we have $\left|A_{\text{aug}}^\pi(s,a)\right| = |A_R^\pi(s,a) + \alpha A_{\text{safe}}^\pi(s,a)| \leq |A_R^\pi(s,a)| + \alpha$. $\square$

We will split $A_{\text{aug}}^\pi$ into two parts $A_R^\pi$ and $A_C^\pi$. Since the ball $B(a, \epsilon)$ is small enough that local averages approximate point values, our construction reduces to

$$A_{\text{safe}}^{\pi_k}(s,a) = Q_{\text{safe}}^{\pi_k}(s,a) - V_{\text{safe}}^{\pi_k}(s) \approx -\frac{Q_C^{\pi_k}(s,a) - V_C^{\pi_k}(s)}{V_C^{\pi_k}(s) + \varepsilon} = -\frac{A_C^{\pi_k}(s,a)}{V_C^{\pi_k}(s) + \varepsilon}. \quad (19)$$

Hence, we have

$$A_{\text{aug}}^{\pi_k}(s,a) = A_R^{\pi_k}(s,a) + \alpha A_{\text{safe}}^{\pi_k}(s,a) \approx A_R^{\pi_k}(s,a) - \lambda(s) A_C^{\pi_k}(s,a), \tag{20}$$

where

$$\lambda(s) = \frac{\alpha}{V_C^{\pi_k}(s) + \epsilon}. \tag{21}$$

We assume that the reward and cost are bounded by $R_{\max}$ and $C_{\max}$, respectively. Then, for all $s, a$, we have

$$|A_R^{\pi_k}(s,a)| \leq \frac{R_{\max}}{1-\gamma}, |A_C^{\pi_k}(s,a)| \leq \frac{C_{\max}}{1-\gamma} \text{ and } V_C^{\pi_k}(s) \leq \frac{C_{\max}}{1-\gamma}. \tag{22}$$

This implies that, for all $s$, we have

$$\lambda(s) = \frac{\alpha}{V_C^{\pi_k}(s) + \epsilon} \geq \frac{\alpha}{\frac{C_{\max}}{1-\gamma} + \epsilon} = \frac{\alpha(1-\gamma)}{C_{\max} + \epsilon(1-\gamma)}. \tag{23}$$

Moreover, by rearranging Equation 20, we have

$$A_C^{\pi_k}(s,a) \approx \frac{A_R^{\pi_k}(s,a) - A_{\text{aug}}^{\pi_k}(s,a)}{\lambda(s)}.$$

Combining Equations 22 and 23, and Lemma 2, we have

$$|A_C^{\pi_k}(s,a)| \leq \frac{1}{\lambda(s)} \left( |A_R^{\pi_k}(s,a)| + |A_{\text{aug}}^{\pi_k}(s,a)| \right) \tag{24}$$

$$\leq \frac{1}{\lambda(s)} \left( \frac{R_{\max}}{1-\gamma} + \frac{R_{\max}}{1-\gamma} + \alpha \right) \tag{25}$$

$$\leq \frac{C_{\max} + \epsilon(1-\gamma)}{\alpha(1-\gamma)} \left( \frac{2R_{\max}}{1-\gamma} + \alpha \right) \tag{26}$$

$$= \frac{2R_{\max}(C_{\max} + \epsilon(1-\gamma))}{\alpha(1-\gamma)^2} + \frac{C_{\max}}{(1-\gamma)} + \epsilon. \tag{27}$$

Thus, we have

$$|A_C^{\pi_k}(s,a)| \leq O\left(\frac{1}{\alpha}\right) + \frac{C_{\max}}{(1-\gamma)} + \epsilon. \tag{28}$$

By the definition of $\epsilon_C^{\pi_{k+1}} = \max_s \left| \mathbb{E}_{a \sim \pi_{k+1}}[A_C^{\pi_k}(s,a)] \right|$, and Equation 28, CPO worst-case bound (Proposition 2 in Achiam et al. (2017))

$$J_C(\pi_{k+1}) \leq J_C(\pi_k) + \frac{\sqrt{2\delta}\gamma}{(1-\gamma)^2} \varepsilon_C^{\pi_{k+1}} \tag{29}$$

becomes

$$J_C(\pi_{k+1}) \leq J_C(\pi_k) + O\left(\frac{\sqrt{\delta}}{\alpha}\right) + \frac{C_{\max}\sqrt{2\delta\gamma}}{(1-\gamma)^3} + \epsilon \frac{\sqrt{2\delta}\gamma}{(1-\gamma)^2}, \tag{30}$$

showing that stronger safety regularization (larger $\alpha$) tightens the worst-case constraint-violation bound for a fixed KL radius $\delta$.

## C    EXECUTION-TIME EFFICIENCY

| Robots | Methods & Metrics | Goal | Button | Push | Circle |
|--------|-------------------|------|--------|------|--------|
| Car | RCPO Runtime (s) | 3.94±0.03 | 4.18±0.04 | 4.72±0.01 | 2.58±0.02 |
|  | Ours Runtime (s) | 5.05±0.09 | 5.63±0.10 | 6.54±0.13 | 3.04±0.03 |
|  | Ours Shield Triggers (%) | 18.10±2.16 | 27.51±2.24 | 25.72±1.37 | 7.32±2.95 |
| Point | RCPO Runtime (s) | 3.81±0.01 | 4.14±0.03 | 3.48±0.01 | 2.49±0.16 |
|  | Ours Runtime (s) | 4.32±0.66 | 5.60±0.11 | 5.03±0.21 | 2.61±0.04 |
|  | Ours Shield Triggers (%) | 15.51±3.82 | 29.07±2.49 | 7.53±0.54 | 7.25±2.84 |

Table 1: Runtime (in seconds) and shielding rate (in percent) across tasks for each robot type. "Ours" refers to the combined method using SRO and the adaptive shield.

Table 1 compares the average runtime per episode between the baseline method (RCPO) and our full approach (SRO + Shield). Across all tasks and both robot types, our method introduces only a modest runtime overhead, demonstrating practical efficiency during execution. Additionally, the shield trigger rate remains moderate, indicating that safety shielding is invoked selectively and does not dominate execution time.

For runtime execution comparisons, we standardize the hardware to ensure fairness. Experiments are run on a CPU server with 256 GB of memory, dual AMD CPUs (56 cores, 224 threads).

# D   PSEUDOCODE FOR THE SAFETY-REGULARIZED ACTOR-CRITIC OBJECTIVE.

For clarity in this section only, we denote the parameters for policy, reward value critic, cost value critic, and cost Q-critic by $\theta$, $\phi$, $\psi$, and $\omega$, respectively. We use $sg(\cdot)$ to indicate a stop-gradient operation.

---

**Algorithm 1** Actor-Critic with Safety-Regularized Objective

---

1: **Initialize:** Policy $\pi_\theta$, Reward Critic $V_R^\phi$, Cost Value Critic $V_C^\psi$, Cost Q-Critic $Q_C^\omega$.
2: **Hyperparameters:** Safety bonus coefficient $\alpha$, KL constraint $\delta_{KL}$.
3: **for** each training epoch **do**
4:     Collect trajectories $\mathcal{D}$ using current policy $\pi_\theta$.
5:     Compute targets $V_R^{\text{targ}}, V_C^{\text{targ}}$ and advantages $A_R, A_C$ using GAE.
6:     Store observations, actions, log-probs, and targets in buffer.
7:     **for** $k = 1$ to $K$ update iterations **do**
8:         Sample mini-batch $\mathcal{B} = \{(s, a, \log \pi_{old}, V^{\text{targ}}, A)\}$ from $\mathcal{D}$.
9:         **// 1. Update Reward Critic**
10:         Minimize $\mathcal{L}_{V_R} = \mathbb{E}_\mathcal{B}\left[(V_R^\phi(s) - V_R^{\text{targ}})^2\right]$.
11:         **// 2. Update Cost Critics ($V_C$ and $Q_C$)**
12:         Compute $V_{stop\_grad} = sg(V_C^\psi(s))$.
13:         Minimize $\mathcal{L}_{V_C} = \mathbb{E}_\mathcal{B}\left[(V_{stop\_grad} - V_C^{\text{targ}})^2\right]$.
14:         Minimize $\mathcal{L}_{Q_C} = \mathbb{E}_\mathcal{B}\left[((Q_C^\omega(s, a) - V_{stop\_grad}) - A_C)^2\right]$.
15:         **// 3. Compute Safety Estimates**
16:         Sample $N$ noise vectors $\epsilon \sim \mathcal{N}(0, \sigma)$ to generate local actions $a' = a + \epsilon$.
17:         Estimate local Cost Q-values $Q_{approx} = \frac{1}{N}\sum_{a'} \pi(a' \mid s)Q_C^\omega(s, a')$.
18:         Compute safety regularizer $Q_{\text{safe}} = \frac{Q_{approxi}}{sg(V_C^\psi)+\epsilon} \approx$ Equation 6, .
19:         **// 4. Update Actor**
20:         Compute surrogate advantage:

$$A_{\text{aug}} = A_R + \alpha \cdot Q_{\text{safe}}$$

21:         Update $\theta$ by maximizing policy objective $\mathcal{L}_\pi$ using $A_{\text{aug}}$.
22:         **// 5. Early Stopping**
23:         **if** $D_{KL}(\pi_{\theta_{old}}||\pi_\theta) > \delta_{KL}$ **then**
24:             Break inner loop.
25:         **end if**
26:     **end for**
27: **end for**

---

# E   CONNECTION TO CONTROL THEORY

In this section, we introduce a brief overview of barrier certificate approaches in control theory and then relate these ideas to our shielding mechanism.

Recent advances in safe control and safe reinforcement learning suggest using Control Barrier Functions (CBFs) or barrier-like certificates to establish forward invariance of a safe set. Classical and neural CBF methods construct a differentiable barrier function $h(s)$ whose evolution satisfies a discrete- or continuous-time invariance condition, and use this certificate to guarantee safety under learned or partially known dynamics. This paradigm has been widely applied to safe RL, including disturbance-observer-based barrier methods (Cheng et al., 2023), reachability-based approximations (Ganai et al., 2023), soft-barrier formulations for stochastic environments (Wang et al., 2023), neural CBFs integrated directly into RL pipelines (Xiao et al., 2023), and the joint use of CBFs and control Lyapunov functions (CLFs) for enhanced stability and safety under model uncertainty (Choi et al., 2020). These approaches learn or optimize a barrier function jointly with the policy or the dynamics model, and safety depends on the existence of a valid barrier certificate, which is often difficult to find.

**Control Barrier Function**   A function $\alpha : [0, a) \to [0, \infty)$ is called a *class-$\mathcal{K}$ function* if it is continuous, strictly increasing, and satisfies $\alpha(0) = 0$. In classical nonlinear control, forward invariance of a safe set $\mathcal{S}_{\text{safe}} \subseteq \mathcal{S}$ is certified through a Control Barrier Function (CBF) $h : \mathcal{S} \to \mathbb{R}$ that satisfies the differential constraint

$$\dot{h}(s) + \alpha(h(s)) \geq 0, \tag{31}$$

for an extended class-$\mathcal{K}$ function $\alpha$. If (31) holds for all admissible controls, then trajectories starting in $\mathcal{S}_{\text{safe}} = \{s \mid h(s) \geq 0\}$ remain in that set for all future times, guaranteeing forward invariance. The shielding guarantee proposed in Theorem 1 is closely related to the notion of *forward invariance* in control theory.

**Relation to our Shield.**   Our adaptive shield provides an analogous guarantee in the discrete-time and data-driven setting. Rather than enforcing the differential condition (31), we bound the change in a safety function $\nu(e(s_t), E_t)$ between successive steps using its Lipschitz continuity,

$$\nu(e(s_{t+1}), E_{t+1}) \geq \nu(e(s_t), E_t) - L_\nu \left\| e(s_{t+1}) - e(s_t) \right\| \geq \nu(e(s_t), E_t) - L_\nu \Delta_{\max}. \tag{32}$$

Thus, whenever $\nu(e(s_t), E_t) > L_\nu \Delta_{\max}$, the next state $s_{t+1}$ remains within the safe region. This is a discrete-time forward-invariance condition derived from the structure of the cost function $C(s_t, a_t, s_{t+1})$ and the Lipschitz property of $\nu$. The adaptive conformal bound $\Gamma_t$ introduced in the shield plays the role of a stochastic disturbance margin, producing a probabilistic forward invariance guarantee.

While neural and classical CBF methods construct a barrier certificate $h(s)$ satisfying the invariance condition Equation 31 either through analytical dynamics or by learning $h$ jointly with a dynamics model, our approach leverage the structure of the cost function such as Lipschitz property, and conformal prediction bound without learning a barrier function.

## F  ABLATION STUDY ON SAFETY BONUS AND SAMPLING SIZE

We evaluate the hyperparameter sensitivity of our method combined with RCPO, focusing on safety bonus $\alpha$ and sampling size $s$.

**Varying Safety Bonus $\alpha$ with RCPO.** To assess the sensitivity of the safety bonus, we vary the safety bonus $\alpha$ across $\{0.05, 0.1, 0.5, 1.0\}$. We observe that performance with SRO often improves both reward and safety. This is mainly because SRO encourages the policy to select safe actions while also exploring under-explored actions. However, SRO does not consistently enhance both reward and safety; instead, it frequently improves either reward or cost. These findings align with the theoretical results presented in Appendix B.

| Env. Algo. | Point-Goal R ↑ | C(%) ↓ | Point-Button R ↑ | C(%) ↓ | Point-Push R ↑ | C(%) ↓ | Point-Circle R ↑ | C(%) ↓ |
|---|---|---|---|---|---|---|---|---|
| RCPO | 16.35±1.14 | 2.50±0.02 | 8.97±0.56 | 1.69±0.23 | 0.12±0.15 | 0.63±0.28 | **30.78±5.55** | 2.47±0.51 |
| SRO + RCPO (α=0.05) | 17.49±1.82 | 2.89±0.52 | 10.38±1.89 | 1.76±0.24 | **0.33±0.16** | 0.88±0.48 | 24.33±4.58 | 1.55±0.03 |
| SRO + RCPO (α=0.1) | 17.56±1.35 | 2.63±0.25 | 10.28±2.00 | **1.64±0.23** | 0.32±0.15 | 0.65±0.37 | 25.95±4.37 | **0.98±0.02** |
| SRO + RCPO (α=0.5) | **17.58±0.56** | 2.49±0.34 | **11.26±1.28** | 1.82±0.31 | 0.33±0.12 | **0.61±0.10** | 25.18±4.50 | 1.28±0.29 |
| SRO + RCPO (α=1.0) | 15.93±0.78 | **2.37±0.11** | 9.05±1.43 | 1.77±0.43 | 0.33±0.04 | 0.82±0.56 | 24.97±3.74 | 1.55±0.34 |

| Env. Algo. | Car-Goal R ↑ | C(%) ↓ | Car-Button R ↑ | C(%) ↓ | Car-Push R ↑ | C(%) ↓ | Car-Circle R ↑ | C(%) ↓ |
|---|---|---|---|---|---|---|---|---|
| RCPO | 15.64±2.27 | 2.23±0.33 | 5.74±0.34 | 1.89±0.22 | -0.09±0.13 | 0.60±0.09 | **11.83±0.49** | 1.58±0.57 |
| SRO + RCPO (α=0.05) | **18.54±0.65** | 2.13±0.14 | 7.01±0.51 | **1.80±0.26** | -0.10±0.17 | 0.88±0.25 | 11.29±0.28 | **1.29±0.19** |
| SRO + RCPO (α=0.1) | 15.88±1.12 | 2.31±0.66 | **7.68±1.21** | 1.85±0.19 | **0.06±0.09** | 0.68±0.41 | 11.47±0.53 | 1.51±0.61 |
| SRO + RCPO (α=0.5) | 17.33±0.58 | 2.55±0.50 | 6.71±1.11 | 1.88±0.46 | -0.05±0.12 | **0.59±0.39** | 11.19±0.25 | 1.52±0.47 |
| SRO + RCPO (α=1.0) | 16.73±2.46 | **1.73±0.29** | 7.89±0.95 | 1.95±0.47 | -0.14±0.18 | 0.67±0.19 | 11.50±0.16 | 2.15±0.50 |

Table 2: Ablation Study on the Varying Effects of Safety Bonus $\alpha$ on Safety and Performance. Best performances (highest return and lowest cost rate) are highlighted in bold.

**Varying Sampling Numbers $s$ with RCPO.** We evaluate the impact of sampling size, varying it across $\{5, 10, 20, 50\}$ by fixing safety bonus $\alpha = 1.0$. Table 3 demonstrates that sampling size influences performance. Sampling numbers exhibit no consistent pattern due to randomness in the sampling procedure. This arises primarily from high prediction errors, which often lead to incorrect action sampling, even within conformal prediction boundaries. For example, a large error widens the conformal interval range, causing the shield to include numerous sampled actions to meet the probabilistic guarantee. However, even when selecting actions based on safety scores, these high errors may inaccurately represent safe actions.

| Env. Algo. | Point-Goal R ↑ | C(%) ↓ | Point-Button R ↑ | C(%) ↓ | Point-Push R ↑ | C(%) ↓ | Point-Circle R ↑ | C(%) ↓ |
|---|---|---|---|---|---|---|---|---|
| RCPO | 16.35±1.14 | 2.50±0.02 | **8.97±0.56** | 1.69±0.23 | 0.12±0.15 | 0.63±0.28 | **30.78±5.55** | 2.47±0.51 |
| Shield + SRO (s=5) | 12.83±1.91 | 2.27±0.60 | 8.68±2.51 | 2.43±0.39 | 0.13±0.27 | **0.51±0.27** | 26.76±4.71 | **1.38±0.45** |
| Shield + SRO (s=10) | **16.40±0.73** | 2.29±0.21 | 8.22±3.31 | 1.98±0.72 | **0.36±0.40** | 0.65±0.04 | 29.97±0.43 | 2.16±0.43 |
| Shield + SRO (s=20) | 14.74±1.18 | **1.96±0.37** | 7.43±1.55 | 2.19±0.45 | 0.03±0.18 | 0.90±0.53 | 26.76±5.40 | 1.81±0.36 |
| Shield + SRO (s=50) | 14.37±0.65 | 2.34±0.23 | 7.34±1.27 | **1.58±0.47** | 0.31±0.20 | 0.53±0.37 | 28.24±4.54 | 1.87±0.63 |

| Env. Algo. | Car-Goal R ↑ | C(%) ↓ | Car-Button R ↑ | C(%) ↓ | Car-Push R ↑ | C(%) ↓ | Car-Circle R ↑ | C(%) ↓ |
|---|---|---|---|---|---|---|---|---|
| RCPO | 15.64±2.27 | 2.23±0.33 | **5.74±0.34** | 1.89±0.22 | -0.09±0.13 | 0.60±0.09 | **11.83±0.49** | 1.58±0.57 |
| Shield + SRO (s=5) | 14.43±0.27 | 2.12±0.15 | 5.24±0.90 | 1.86±0.28 | 0.09±0.07 | 0.75±0.24 | 11.55±0.29 | **1.28±1.02** |
| Shield + SRO (s=10) | 13.65±0.35 | **1.84±0.14** | 5.52±0.90 | 1.92±0.43 | **0.23±0.09** | 0.86±0.24 | 11.61±0.67 | 1.44±0.84 |
| Shield + SRO (s=20) | **15.69±0.47** | 2.51±0.50 | 4.93±0.81 | 1.99±0.20 | 0.04±0.18 | 0.61±0.41 | 11.44±0.20 | 1.88±1.02 |
| Shield + SRO (s=50) | 13.23±1.73 | 2.65±0.86 | 4.88±1.37 | **1.52±0.17** | 0.00±0.10 | **0.50±0.20** | 11.08±0.37 | 1.49±0.74 |

Table 3: Ablation Study on the Varying Effects of Sampling Numbers $s$ on Safety and Performance with fixed safety bonus $\alpha = 1.0$. Best performances (highest return and lowest cost rate) are highlighted in bold.

# G ADAPTIVE SHIELDING AND SAFETY-REGULARIZED OBJECTIVE WITH PPO-LAG

Our method is compatible with a wide range of RL algorithms, as it wraps the policy with a shielding layer and incorporates an augmented term based on $Q_C$ and $V_C$, which are commonly used in safe RL algorithms. To demonstrate this, we examine its impact when applied to a PPO-Lagrangian-based policy. Here, we study PPO-Lag augmented with SRO and Adaptive Shielding mechanisms. The baseline PPO-Lag method is provided with access to the hidden parameter $\phi$. First, we analyze how SRO affects PPO-Lagrangian method. Then, we show how Shielding mechanism combined with SRO affects PPO-Lagrangian method. All results shown represent the mean reward and cost rate over the last 20 epochs of training across seeds.

**Safety-Regularized Objective with PPO-Lag.** Table 4 demonstrate that SRO generally enhances safety. In Point Robot case, a clear pattern emerges: higher safety bonus values $\alpha$ improve safety the most, while lower $\alpha$ have minor effects. Notably, adding SRO does not degrade reward performance substantially, with cost violations improving by up to 20% (Point-Button) to as much as 520% (Point-Circle). Meanwhile, reward degradation occurs only in the Point-Circle and Car-Button environments; in many other cases, SRO improves not only safety but also the reward signal. This is primarily because our augmented objective is bounded in $(-1, 0]$, mildly influencing the training objective to compensate for actions leading to zero long-term cost violations and under-explored actions, without causing significant shifts during training due to the bounded values.

| Env. Algo. | Point-Goal | | Point-Button | | Point-Push | | Point-Circle | |
|---|---|---|---|---|---|---|---|---|
| | R ↑ | C(%) ↓ | R ↑ | C(%) ↓ | R ↑ | C(%) ↓ | R ↑ | C(%) ↓ |
| PPOLag | 18.20±0.78 | 3.35±0.20 | 10.83±1.56 | 2.94±0.22 | 0.24±0.19 | 1.15±0.27 | **30.90±3.52** | 13.38±6.74 |
| SRO + PPOLag ($\alpha$=0.05) | **19.41±0.45** | 2.97±0.25 | 10.92±1.65 | 3.16±0.54 | 0.19±0.08 | 1.16±0.10 | 25.20±2.31 | 4.56±1.87 |
| SRO + PPOLag ($\alpha$=0.1) | 17.65±0.26 | 2.52±0.07 | **11.63±0.80** | 3.04±0.58 | **0.53±0.27** | 1.32±0.36 | 27.79±1.04 | 2.93±1.49 |
| SRO + PPOLag ($\alpha$=0.5) | 17.12±0.97 | **2.51±0.50** | 10.49±0.68 | 2.88±0.32 | 0.32±0.03 | 0.85±0.24 | 26.50±3.41 | 3.58±1.07 |
| SRO + PPOLag ($\alpha$=1.0) | 16.98±1.36 | 2.79±0.36 | 9.69±0.55 | **2.44±0.35** | 0.36±0.20 | **0.59±0.16** | 26.07±4.88 | **2.15±1.27** |

| Env. Algo. | Car-Goal | | Car-Button | | Car-Push | | Car-Circle | |
|---|---|---|---|---|---|---|---|---|
| | R ↑ | C(%) ↓ | R ↑ | C(%) ↓ | R ↑ | C(%) ↓ | R ↑ | C(%) ↓ |
| PPOLag | 15.85±1.01 | 3.36±0.46 | **8.62±1.26** | 3.58±0.36 | **0.02±0.09** | 1.14±0.12 | 11.48±0.38 | 2.57±0.23 |
| SRO + PPOLag ($\alpha$=0.05) | 16.10±1.07 | **3.08±0.11** | 8.37±1.69 | 3.31±1.13 | 0.02±0.07 | **0.95±0.20** | 11.24±0.13 | 2.32±0.36 |
| SRO + PPOLag ($\alpha$=0.1) | **16.81±1.17** | 3.16±0.18 | 8.53±1.64 | 3.56±0.36 | 0.02±0.04 | 1.07±0.09 | 11.46±0.45 | 2.00±0.20 |
| SRO + PPOLag ($\alpha$=0.5) | 15.50±1.11 | 3.18±0.34 | 8.24±0.54 | **3.14±0.43** | -0.02±0.10 | 1.14±0.29 | **11.52±0.47** | 2.02±0.88 |
| SRO + PPOLag ($\alpha$=1.0) | 16.41±0.57 | 3.60±0.53 | 8.11±1.17 | 3.78±0.16 | -0.01±0.10 | 0.99±0.23 | 11.42±0.56 | **1.74±0.60** |

Table 4: Ablation Study on the Varying Effects of Safety Bonus $\alpha$ on Safety and Performance. Best performances (highest return and lowest cost rate) are highlighted in bold.

**Adaptive Shielding with PPO-Lag.** For sampling numbers, unlike the safety bonus $\alpha$, no consistent pattern emerges due to randomness in the sampling procedure. The same reasoning outlined in Appendix F applies here. Thus, reducing prediction errors and mitigating the inherent randomness in the sampling process represent key areas for future research to enhance shielding-based approaches.

| Env. Algo. | Point-Goal | | Point-Button | | Point-Push | | Point-Circle | |
|---|---|---|---|---|---|---|---|---|
| | R ↑ | C(%) ↓ | R ↑ | C(%) ↓ | R ↑ | C(%) ↓ | R ↑ | C(%) ↓ |
| PPOLag | 18.20±0.78 | 3.35±0.20 | 10.83±1.56 | 2.94±0.22 | 0.24±0.19 | 1.15±0.27 | **30.90±3.52** | 13.38±6.74 |
| Shield + SRO (s=5) | 17.41±1.96 | 2.84±0.36 | 10.27±1.30 | 2.87±0.36 | **0.38±0.11** | 1.01±0.12 | 25.74±3.26 | 4.25±1.30 |
| Shield + SRO (s=10) | **18.38±0.84** | 2.86±0.25 | 10.18±0.86 | 2.87±0.23 | 0.20±0.06 | 0.92±0.36 | 27.34±8.09 | 2.42±2.48 |
| Shield + SRO (s=20) | 17.10±2.28 | 2.75±0.32 | 8.84±0.05 | **2.67±0.27** | 0.15±0.12 | **0.82±0.10** | 28.83±3.36 | **2.03±0.87** |
| Shield + SRO (s=50) | 17.44±0.90 | **2.60±0.26** | **11.23±3.43** | 3.12±1.07 | 0.37±0.10 | 1.27±0.64 | 27.55±4.32 | 2.43±1.54 |

| Env.
Algo. | Car-Goal | | Car-Button | | Car-Push | | Car-Circle | |
|---|---|---|---|---|---|---|---|---|
| | R↑ | C(%)↓ | R↑ | C(%)↓ | R↑ | C(%)↓ | R↑ | C(%)↓ |
| PPOLag | 15.85±1.01 | **3.36±0.46** | **8.62±1.26** | 3.58±0.36 | 0.02±0.09 | 1.14±0.12 | **11.48±0.38** | 2.57±0.23 |
| Shield + SRO (s=5) | 15.56±0.47 | 3.66±0.64 | 7.57±0.39 | 3.28±0.12 | 0.03±0.11 | 1.48±0.46 | 10.38±0.39 | 1.43±0.70 |
| Shield + SRO (s=10) | **16.87±5.59** | 3.79±0.24 | 7.62±0.91 | 3.35±0.57 | 0.05±0.10 | 1.12±0.30 | 10.15±0.67 | 1.01±0.27 |
| Shield + SRO (s=20) | 15.97±1.36 | 3.43±0.83 | 7.56±1.01 | **2.64±0.47** | **0.09±0.14** | 1.22±0.36 | 10.33±0.15 | **0.93±0.09** |
| Shield + SRO (s=50) | 13.93±2.09 | 3.48±0.50 | 6.25±1.47 | 3.22±0.41 | 0.05±0.10 | **1.01±0.55** | 11.38±0.54 | 1.49±0.07 |

Table 5: Ablation Study on the Varying Effects of Sampling Numbers $s$ on Safety and Performance with fixed safety bonus $\alpha = 1.0$. Best performances (highest return and lowest cost rate) are highlighted in bold.

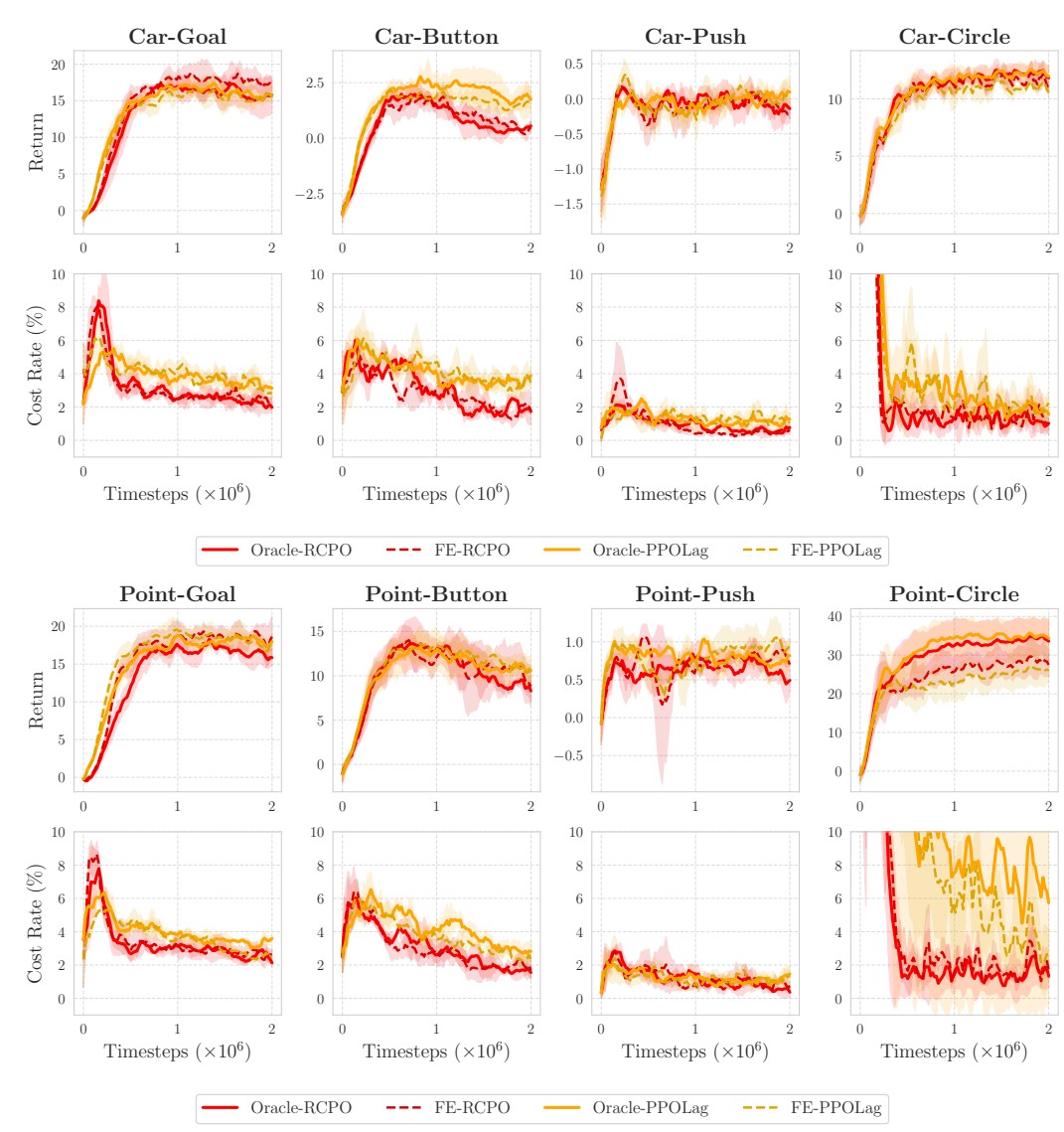

Figure 3: Ablation study on Representation. "Oracle-" refers to a policy directly informed of hidden parameters, while "FE-" denotes the function encoder's representation derived from observations.

## H   FUNCTION ENCODERS AND ITS REPRESENTATION

For completeness, we present the necessary background on function encoders. For more information, see (Ingebrand et al., 2024a; 2025).

**Overview of Function Encoders.** Function encoders, as introduced by Ingebrand et al. (2025), provide a principled framework for representing tasks in a Hilbert space $\mathcal{H}$ through a finite set of neural network-based basis functions $\{g_1, \ldots, g_k\}$, each parameterized by $\theta_j$. A task function $f \in \mathcal{H}$ is approximated as a linear combination of these basis functions:

$$f(x) = \sum_{j=1}^{k} b_j g_j(x \mid \theta_j),$$

where $b_j$ are coefficients tailored to the specific task. This approach enables efficient representation of complex functions, such as those encountered in reinforcement learning or classification, by learning a versatile basis that spans the function space. By defining appropriate inner products, function encoders can generalize to various function spaces, including probability distributions for classification tasks. During supervised training, the neural basis functions $\{g_j\}$ are optimized such that $\sum_j b_j g_j(x_i) \approx f(x_i)$ for all training points $(x_i, f(x_i))$. As expected from functional analysis, using more basis functions increases expressive power and allows modeling more complex dynamics. Empirically, we demonstrate that a larger number of basis functions yields richer representations showing faster convergence of the dynamics prediction loss. Models with fewer basis functions eventually reach similar final performance but require more training epochs (Appendix I).

The training process consists of two phases: offline training of the basis functions and online inference to compute task-specific coefficients. The offline phase optimizes the basis to minimize reconstruction error across a set of source datasets, while the online phase efficiently computes coefficients for new tasks using the learned basis.

**Training Function Encoders via Least Squares.** Function encoder is trained by using a least-squares optimization approach (Ingebrand et al., 2025). Given a set of task functions $\{f_1, \ldots, f_n\}$, the goal is to learn a set of basis functions $\{g_1, \ldots, g_k\}$ parameterized by $\theta$ and these basis functions represent the task functions with varying coefficients $b$. In our implementation, each task function $f_\ell$ is defined by fixing one set of hidden parameters: gravity, mass, damping, density, friction.

The training procedure iteratively minimizes a loss function comprising two components: a reconstruction loss and a regularization term. For each task function $f_\ell$, we compute coefficients $b^\ell = [b_1^\ell, \ldots, b_k^\ell]^T$ that best approximate the target function $f_\ell$ as:

$$b^\ell = \begin{bmatrix} \langle g_1, g_1 \rangle_{\mathcal{H}} & \cdots & \langle g_1, g_k \rangle_{\mathcal{H}} \\ \vdots & \ddots & \vdots \\ \langle g_k, g_1 \rangle_{\mathcal{H}} & \cdots & \langle g_k, g_k \rangle_{\mathcal{H}} \end{bmatrix}^{-1} \begin{bmatrix} \langle f_\ell, g_1 \rangle_{\mathcal{H}} \\ \vdots \\ \langle f_\ell, g_k \rangle_{\mathcal{H}} \end{bmatrix},$$

where $\langle \cdot, \cdot \rangle_{\mathcal{H}}$ denotes the inner product in the Hilbert space, estimated via Monte Carlo integration over collected data points $\{(x_1, f_\ell(x_1)), ((x_2, f_\ell(x_2)), \cdots, ((x_N, f_\ell(x_N))\}$. The reconstructed function is then $\hat{f}_\ell = \sum_{j=1}^{k} b_j^\ell g_j$. The reconstruction loss is defined as:

$$L = \frac{1}{n} \sum_{i=1}^{n} \|f_i - \hat{f}_i\|_{\mathcal{H}}^2,$$

which measures the average squared error between the true and approximated functions. To ensure the basis functions remain well-conditioned, a regularization term is added to the loss function:

$$L_{\text{reg}} = \sum_{i=1}^{k} \left( \|g_i\|_{\mathcal{H}}^2 - 1 \right)^2,$$

which encourages the basis functions to have unit norm. For a learning rate $\alpha$, the parameters $\theta$ are updated via gradient descent: $\theta \leftarrow \theta - \alpha \nabla_\theta (L + L_{\text{reg}})$, until convergence.

**Empirical Evaluation of the Representation.** We investigate the function encoder's representation of varying underlying dynamics $T_\phi$. We evaluate two representations for handling hidden parameters

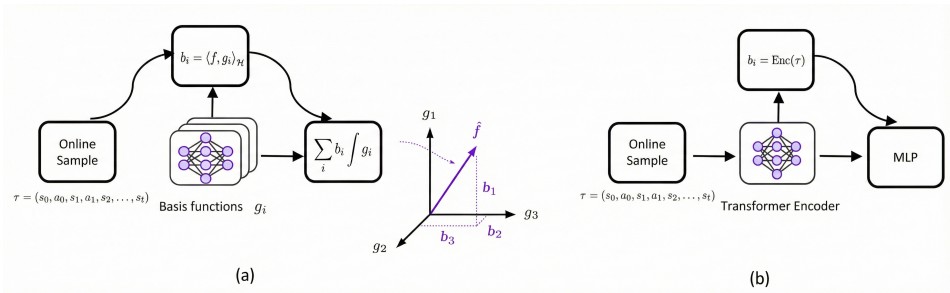

Figure 4: **(a)** Illustration of how function encoders obtain proxy representations of the underlying hidden parameters using online samples. **(b)** A naive approach using transformer encoder to infer hidden parameters from a sequence of online samples. MLP stands for multi-layer perceptron in the Figure.

in our safe RL framework: Oracle representation, where the hidden parameter $\phi$ (a scaling factor for environmental dynamics such as density and damping) is directly provided to the policy by concatenating it with the state input, and Function Encoder (FE) representation, which uses a function encoder $\hat{f}_{\mathrm{FE}}$ to infer the underlying dynamics $T_\phi$, with coefficients of pretrained basis functions serving as the representation. These representations are tested to assess the function encoder's ability to adapt to varying dynamics in Safety Gymnasium tasks. Regarding training hyperparameters, we employed three number of basis functions for the function encoder, trained over 1000 epochs on a dataset of 1000 episodes. Batch size was set to 256. Figure 3 show that the function encoder's representation is often comparable to the oracle representation and, in some cases, outperforms it. The function encoder leverages neural basis functions to represent the space of varying dynamics $\{T_\phi\}_{\phi\in\Phi}$. For instance, just as the $\mathbb{R}^2$ plane is spanned by linear combinations of basis vectors $(0, 1)$ and $(1, 0)$, the dynamics space is captured by neural basis functions, making their coefficients highly informative. This representation often transitions smoothly, as shown in (Ingebrand et al., 2024b), promoting policy effective adaptation to dynamic changes. Consequently, our function encoder's representation frequently matches or surpasses oracle representation performance.

# I  SHIELDING WITH ALTERNATIVE DYNAMICS PREDICTORS.

In this section, we present ablation studies on different dynamics predictors by replacing the function encoder with alternative prediction models. We consider three predictors: a naive transformer-based dynamics model, a probabilistic ensemble model (PEM), and a multilayer perceptron (MLP). Since PEM and MLP are not designed to infer hidden environment parameters $\phi$ from context alone, we provide them with the true $\phi$ as additional input. Following prior baselines that assume access to the environment parameter $\phi$, we refer to these models as Oracle-PEM and Oracle-MLP.

As another baseline described in Figure 4, we use a transformer encoder to process the current episode's trajectory

$$\tau_n = (s_0, a_0, s_1, a_1, \ldots, s_n)$$

and extract a latent representation $b_\phi = \mathrm{Enc}(\tau_n)$ as a proxy for the hidden parameters $\phi$. For a fair comparison to function encoder, the transformer encoder is given 100 samples to infer $b_\phi$, which is then concatenated to the state $s_t$ and fed into the predictor to generate $\hat{s}_t = (s_t, b_\phi)$. All dynamics predictor are designed to hold a comparable parameter numbers (270k-280k). All dynamics predictors are trained on $1,000$ in-distribution episodes and evaluated on 200 out-of-distribution test episodes, which are not used during training.

We first report next state prediction performance on evaluation OOD dataset across all dynamics predictors. We then compare the naive transformer combined with Shield and Shield + SRO against the function encoder combined with the same shielding mechanisms. Note that SRO alone does not use dynamics prediction at deployment.

Figure 5 shows that the function encoder's next-state prediction accuracy closely matches

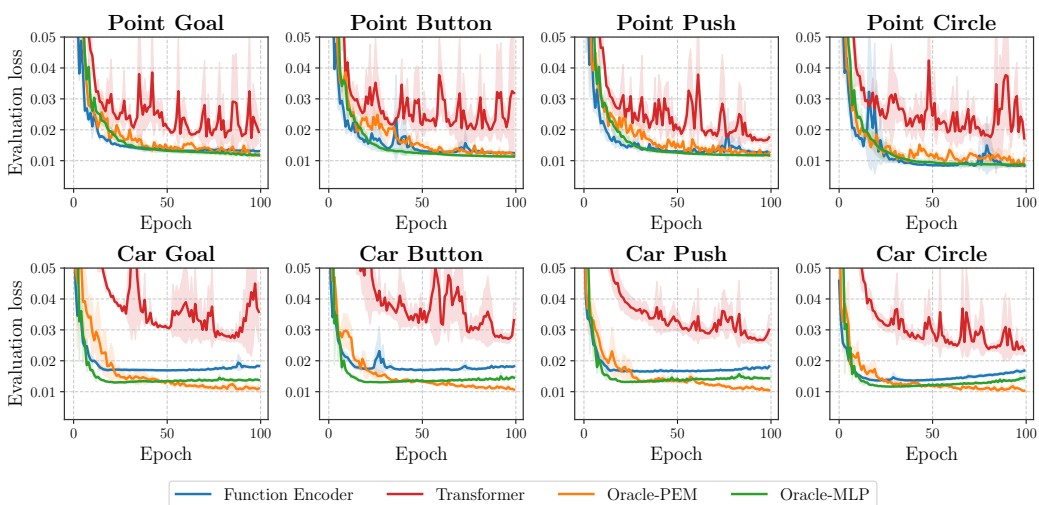

Figure 5: Ablation study evaluating the performance of various dynamics predictors in forecasting the next state. The $y$-axis denotes the average per-sample $\ell_1$-norm error between the true and predicted next states on the test dataset.

that of Oracle-PEM and Oracle-MLP. This observation is consistent with prior findings Ingebrand et al. (2024b;a), which also report that function-encoder-based models can approximate the performance of oracle-informed predictors.

Instead of inferring the hidden parameters $\phi$ through neural basis functions, the Transformer encodes the observation sequence $\tau = (s_0, a_0, s_1, \ldots, s_n)$ and projects the final embedding into a low-dimensional parameter estimate $b_\phi$. Since the true hidden parameters $\phi$'s dimension in safe-navigation domain is 4 (damping, mass, inertia, friction), we evaluate projection dimensions of 3, 6, 9, and 12.

Figure 6 shows an interesting pattern. For the function encoder, increasing the number of basis functions shows early-epoch convergence because the representation has higher capacity, but all configurations eventually converge to similar accuracy, which aligns with the supervised nature of the objective and dataset limitations. In contrast, the Transformer-based encoder does not exhibit a consistent relationship between projection dimension and prediction quality, and its overall performance is less stable.

Finally, Figure 7 shows that the function encoder consistently yields superior performance, achieving higher rewards and fewer constraint violations. This improvement originates from the combination of $(i)$ a more stable and expressive learned representation and $(ii)$ higher next-state prediction accuracy, both of which enhance shielding framework.

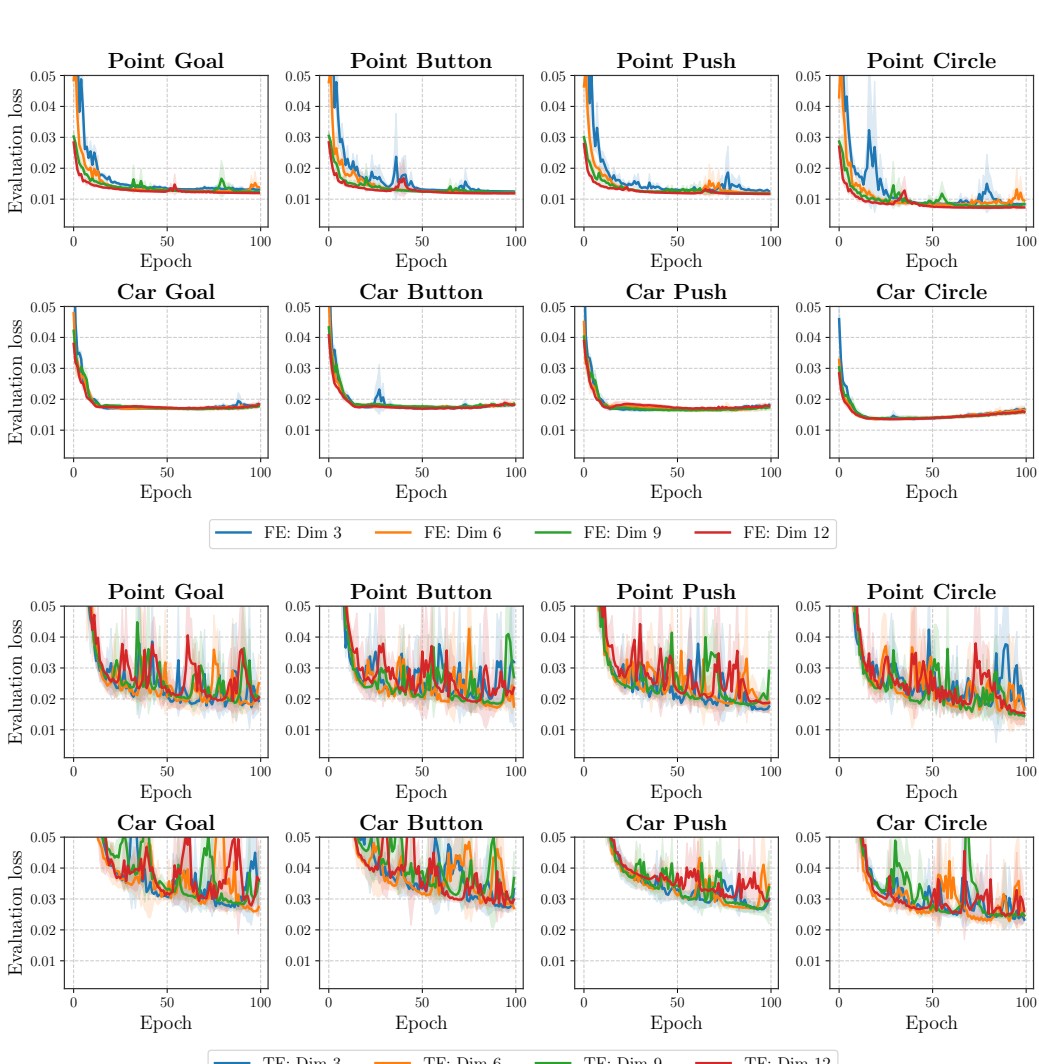

Figure 6: We evaluate how the dimension of the inferred representation $b_\phi$, which is a proxy to hidden parameters $\phi$. We vary the dimension across $\{3, 6, 9, 12\}$, motivated by the ground-truth hidden parameter dimension of $4$ (capturing variations in damping, mass, inertia, and friction). The plot reports the average per-sample $\ell_1$ prediction error between the true and predicted next states on the test set.

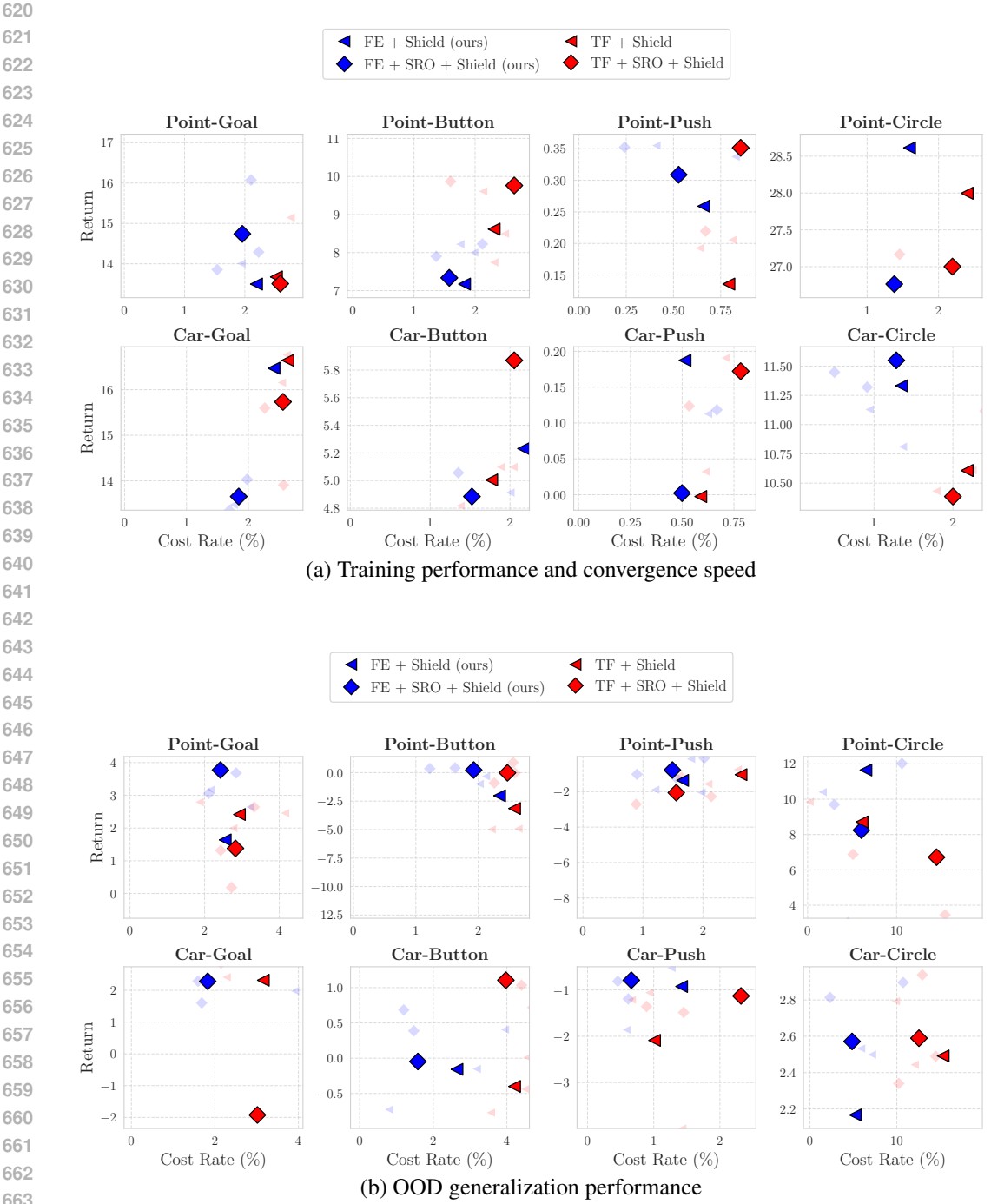

Figure 7: Comparison of our function encoder (FE) representation to a naive Transformer-based (TF) representation when both are used with the proposed shielding framework (all other hyperparameters and conditions remain the same). **(a)** In-distribution training (return vs. cost rate). **(b)** Out-of-distribution (OOD) evaluation (return vs. cost rate). In both settings, our method attains higher returns and fewer constraint violations. This improvement arises from the combination of a more informative latent representation and higher next-state prediction accuracy, both of which strengthen the effectiveness of the shielding mechanism.

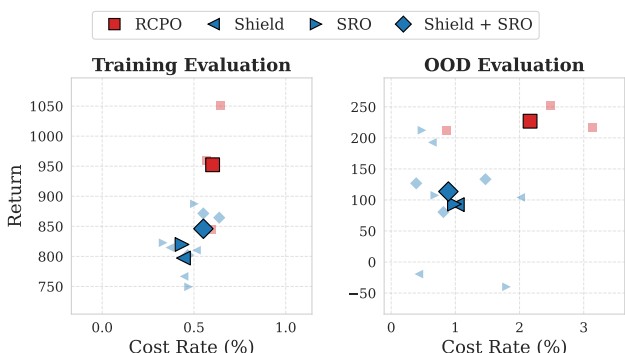

Figure 9: (Left) Return vs. cost rate during the last 20 training epochs, showing the reward–safety tradeoff achieved by each method. (Right) Out-of-distribution (OOD) evaluation performance under shifted hidden parameters.

## J    ADDITIONAL EXPERIMENTS: SAFE VELOCITY CONTROL IN HALFCHEETAH

In this section, we evaluate whether our mechanism generalizes to different robot morphologies and task type by conducting shielding experiments in the HalfCheetah-Velocity task. We follow the safetyconstrained velocity environment from Safety Gymnasium (Ji et al., 2023), but strengthen the safety requirement by tightening the velocity limit from 3.0 to 2.0. As in our earlier settings, the hidden environment parameters vary each episode. To induce richer dynamics variability, we modify the HalfCheetah default parameters, including friction, body segment lengths, and gear ratios with total 14 different hidden parameters:

$$\{friction, torso\_length, bthigh\_length, \cdots, foot\_gear\},$$

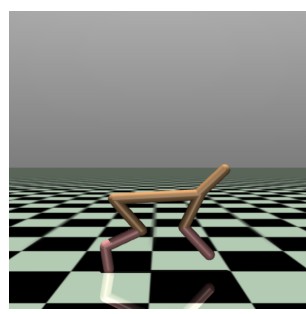

Figure 8: HalfCheetah-Velocity environment.

and resample them at the beginning of each episode. During training, each parameter is scaled uniformly within $[0.7, 1.3]$. For out-of-distribution (OOD) evaluation, we use the disjoint ranges $[0.4, 0.7] \cup [1.3, 1.6]$.

We use 5 neural basis functions in the function encoder, resulting in a 5-dimensional representation that serves as a proxy $b_\phi$ for the hidden parameters $\phi$. As a baseline, RCPO is run with oracle access to the true hidden parameters $\phi$, whereas our method uses the same RCPO implementation without oracle information and instead augment the state with the learned basis coefficients $b_\phi$ as its parameter estimate. All models share the same RL hyperparameters. For hyperparameters for shielding, we use 10 samples of actions, and a safety bonus $\alpha = 1$.

Across both training and OOD settings, Figure 9 shows consistent improvements in safety and overall performance. Our method achieves a favorable reward-cost Pareto frontier compared to the oracle informed baseline, demonstrating that the learned representation and shielding mechanism transfer effectively to more complex robot dynamics.

## K   WHY NOT $Q_C$, BUT $Q_{\text{SAFE}}$?

To effectively guide the policy toward safe behavior, we propose a safety-regularized objective enhanced with $Q_{\text{safe}}$. A natural alternative is to augment the reward value function $Q_R$ with the cost value function $Q_C$, which estimates the expected cost of violating safety constraints, forming $Q_{\text{aug}} = Q_R - \alpha Q_C$. However, this formulation can be transformed into Lagrangian-based safe RL methods, optimizing policies with $Q_R - \lambda Q_C$, where $\lambda$ is a Lagrangian multiplier dynamically adjusted during training. In particular, the Lagrangian multiplier $\lambda$ is updated using a learning rate $lr$. A higher learning rate accelerates the increase of $\lambda$, assigning stronger penalties on the policy for cost violations. $\lambda$ is adjusted by $Q_C \times lr$; larger $Q_C$ or learning rate values lead to faster $\lambda$ growth, which increases the penalty term in the optimization objective $(Q_R - \lambda Q_C)$.

However, Primal-Dual methods such as PPOLag or RCPO are sensitive to the choice of $lr$, often leading to unstable optimization or suboptimal safety-performance trade-offs as shown in Figures 10 and 11. This is because the value of $Q_C$ is highly environment-dependent, varying with the magnitude of costs and the dynamics induced by hidden parameters. In contrast, our safety-regularized objective $Q_{\text{safe}}$ incorporates a normalized term, constrained to $(-1, 0]$. This normalization simplifies controlling the safety bonus by ensuring it remains bounded.

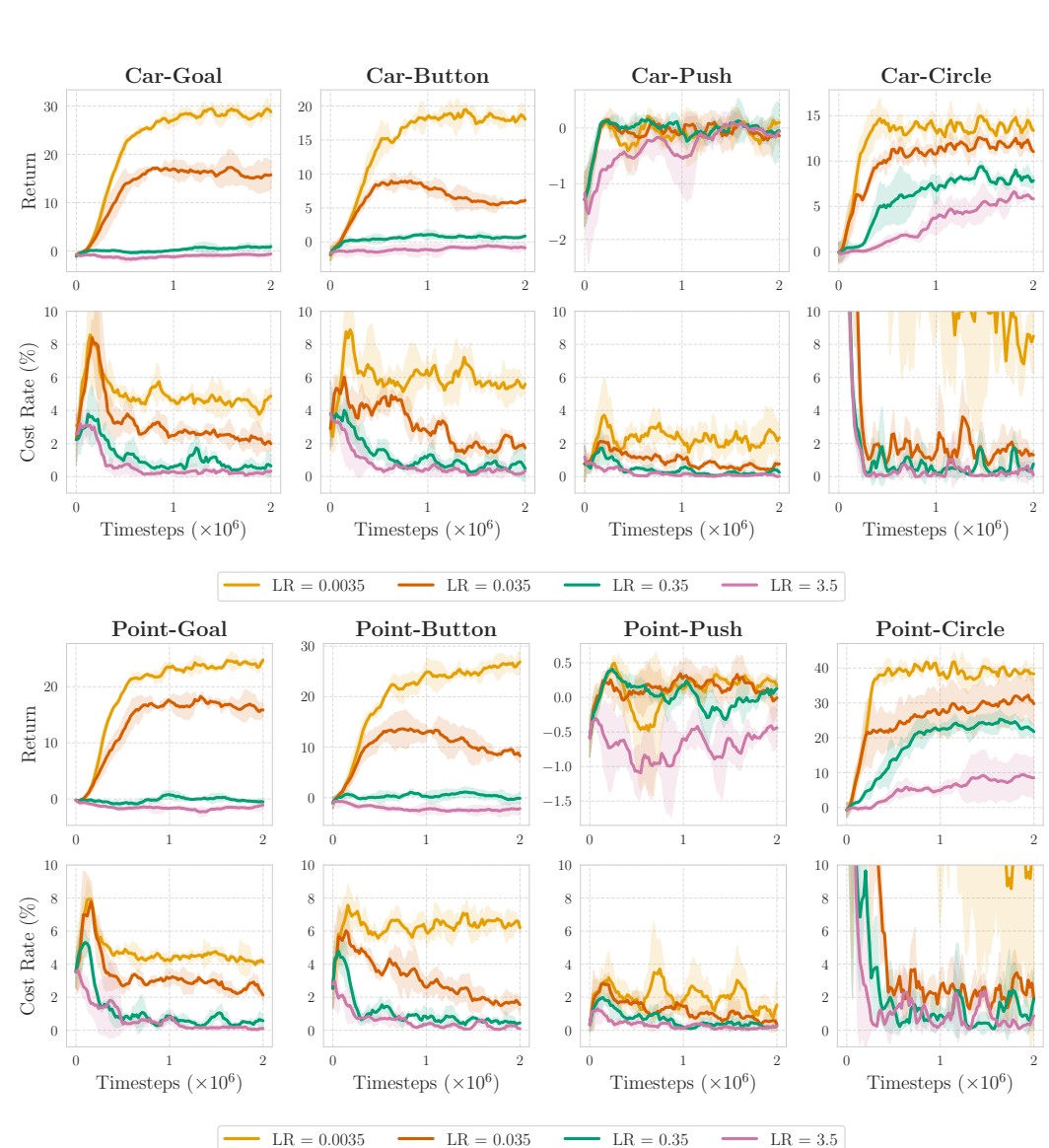

Figure 10: Training curves for RCPO algorithm under varying Lagrangian learning rates. The plots illustrate significant performance variations depending on the learning rate. For our main comparisons, we selected a learning rate of 0.035, which achieves the best trade-off between reward maximization and constraint satisfaction.

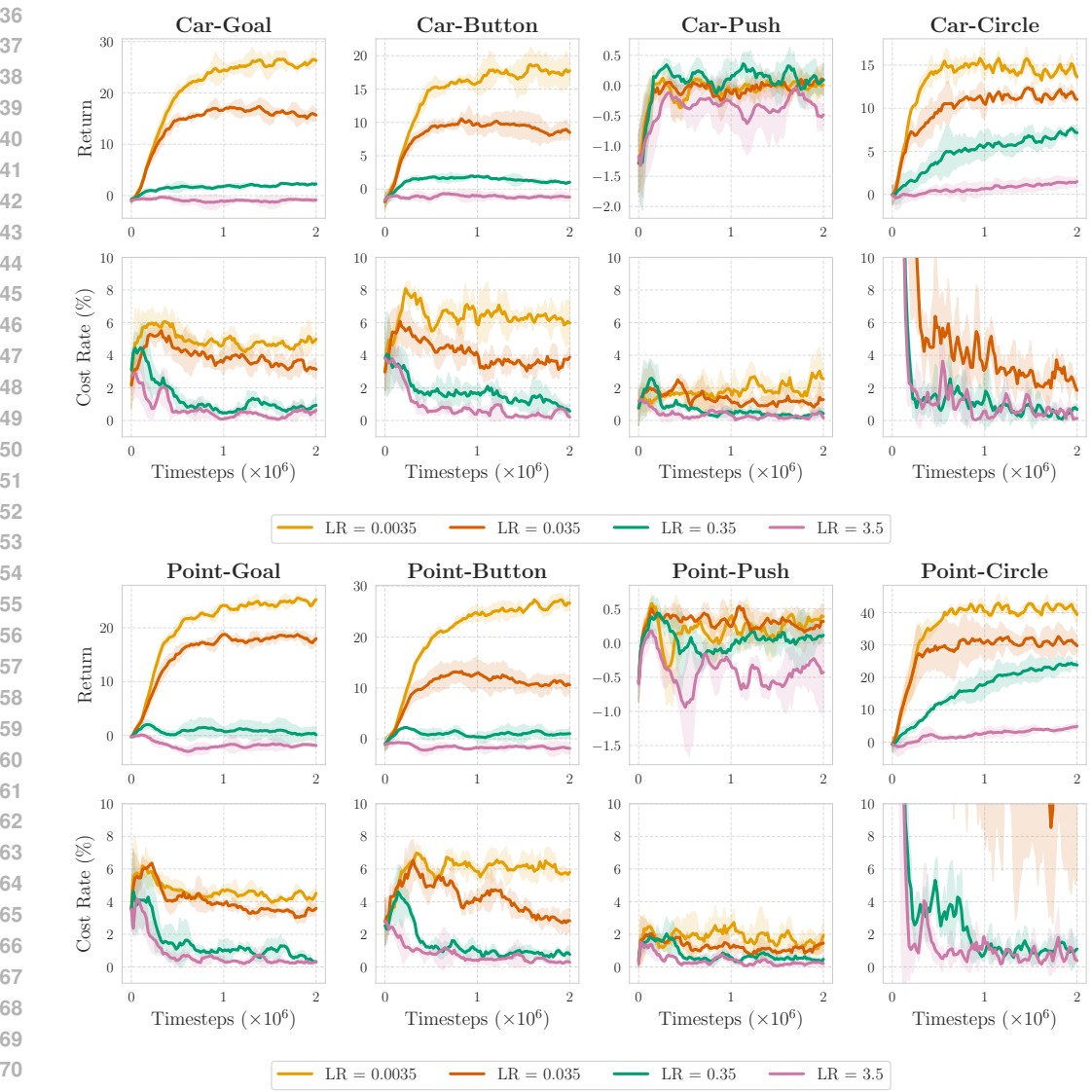

Figure 11: Training curves for PPOLag algorithm under varying Lagrangian learning rates. The plots illustrate significant performance variations depending on the learning rate. For our main comparisons, we selected a learning rate of 0.035, which achieves the best trade-off between reward maximization and constraint satisfaction.

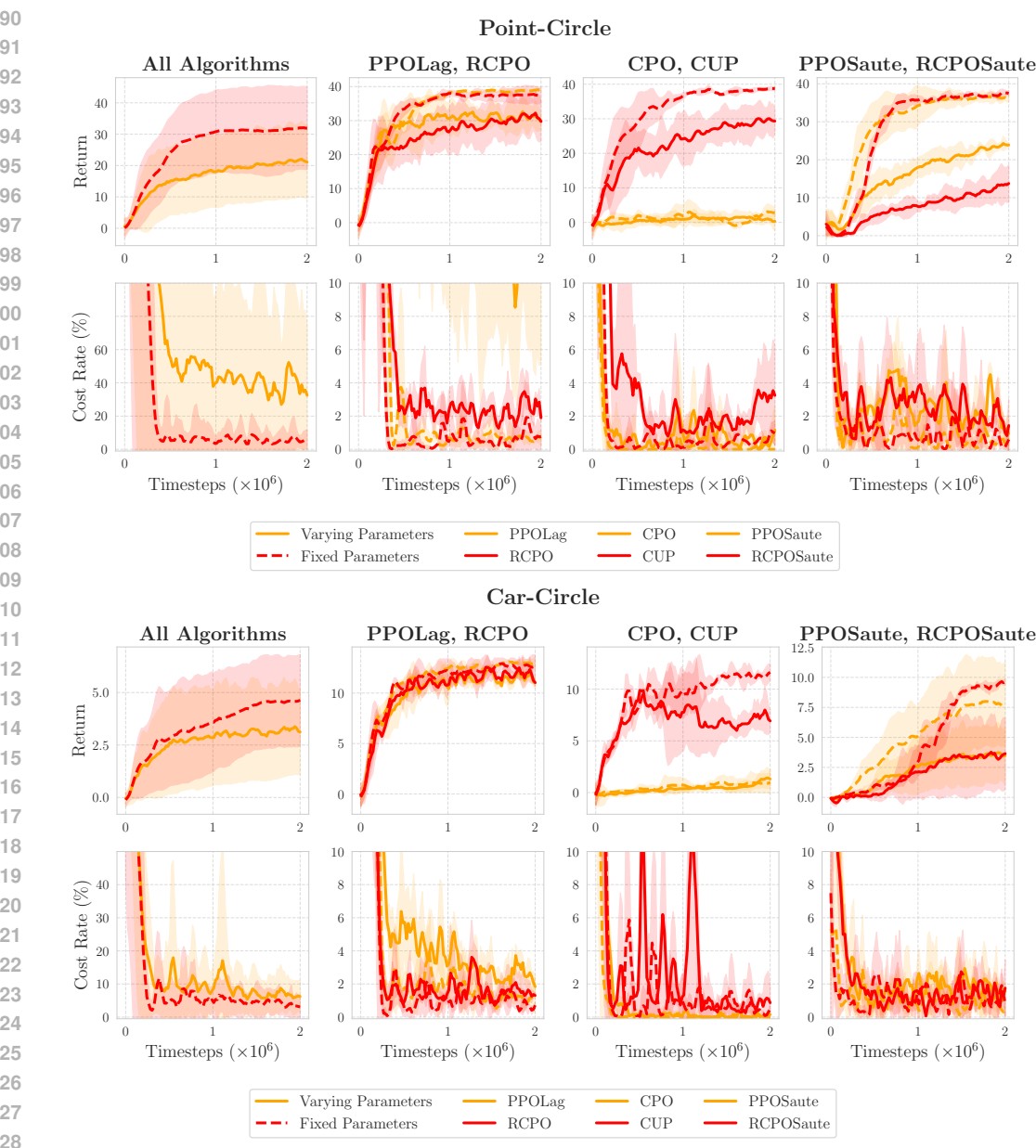

Figure 12: Varying hidden parameters pose significant challenges for safe RL algorithms, even when hidden parameter values are provided as input. The first column plot aggregates results across six algorithms: vanilla Lagrangian methods (RCPO and PPO-Lag), trust region/projection methods (CPO and CUP), and safety augmentation techniques (PPO-Saute and TRPO-Saute). Results with varying hidden parameters are shown as solid lines, while those with fixed parameters are depicted as dotted lines. Columns 1, 2, and 3 present comparative results across these algorithm groups.

## L    FIX PARAMETERS VS. VARYING HIDDEN PARAMETERS

We evaluate the algorithms under two distinct experimental settings to test performance difference when environment dynamics shifts:

- **Fixed Parameters**: In this setting, each algorithm is trained and evaluated in an environment with a single, constant set of parameters $\phi$ (gravity, damping, density, mass, friction) for the entire duration of training.

- **Varying Hidden Parameters**: In contrast, for this setting, the underlying physical parameters $\phi$ of the environment such as gravity, damping, mass, inertia, and friction are randomized at the start of each new episode. To demonstrate the challenge of adapting to varying hidden parameters, we explicitly inform the algorithms of these changes via their input. For example, if gravity is halved from 9.8 to 4.9, a factor of 0.5 is provided as input to the policy.

As demonstrated in Figure 12, we observe a noticeable degradation in the performance of all algorithms under the varying parameter setting, despite being explicitly informed of the magnitude of the changes. More precisely, for Point environment, the total aggregated return across training differs significantly: 25.87 for fixed parameters versus 16.49 for varying parameters. Similarly, for the cost, the values are 18.0 for fixed parameters and 38.41 for varying parameters, reflecting more than double the total cost violations during training.

## M   Cost Functions

In this section, we present two cost functions used in our experiments. Each cost function conforms to the form:

$$C(s_t, a_t, s_{t+1}) = \mathbb{I}\left\{\nu(e(s_{t+1}), E_{t+1}) \leq 0\right\},$$

as defined in Section 4.3, where:

- $e : S \to \mathbb{R}^{n_1}$ extracts agent-centered safety features from the next state $s_{t+1}$,
- $E_{t+1} \in \mathbb{R}^{n_2}$ captures environment features (e.g., obstacle positions, safe region boundaries),
- $\nu : \mathbb{R}^{n_1} \times \mathbb{R}^{n_2} \to \mathbb{R}$ is a Lipschitz continuous function, with $\nu > 0$ indicating safety and $\nu \leq 0$ indicating a violation.

| **Task** | $e(s)$ | $E_t$ | $\nu(e(s), E)$ |
|---|---|---|---|
| Collision avoidance | $\text{pos}(s) \in \mathbb{R}^3$ | $\{X_i\}_{i=1}^M \subset \mathbb{R}^3$ | $\min_i \|e(s) - X_i\|_2 - d_{\text{safe}}$ |
| Safety-region compliance | $\text{pos}(s) \in \mathbb{R}^2$ | $\mathcal{S}_{\text{safe}} \subset \mathbb{R}^2$ | $\text{dist}\big(e(s), \mathbb{R}^2 \backslash \mathcal{S}_{\text{safe}}\big) - \varepsilon$ |

Table 6: Examples of function $\nu$ for different safety tasks.

**Collision Avoidance.** Given the robot's position $\text{pos}(s) \in \mathbb{R}^3$ and the set of obstacle positions $\{X_i\}_{i=1}^M \subset \mathbb{R}^3$ encoded in the state $s$, we mark a transition unsafe whenever the robot comes closer than a safety margin $d > 0$ to any obstacle:

$$C_d(s, a, s') = \mathbb{I}\left[\min_i \|\text{pos}(s) - X_i\| < d\right]$$

Thus $C_d = 1$ whenever the robot violates the distance constraint, encouraging policies that keep a safe distance to obstacles.

**Safety Region Compliance**   To ensure the robot remains within a designated safety region, we evaluate its position in the next state, $\text{pos}(s') = (x, y) \in \mathbb{R}^2$, against a predefined safe region safe_region $\subseteq \mathbb{R}^2$. A penalty is incurred if the position lies outside this region:

$$C_d(s, a, s') = \mathbb{I}[\text{pos}(s') \notin \text{safe\_region}].$$

This cost function assigns a value of 1 when the robot deviates from the safety region, indicating a safety violation.

## N   Average Cost Minimization and Cost Value Function

Our problem formulation targets minimizing the average cost per time step, distinct from the cumulative discounted cost over an infinite horizon typically addressed by Lagrangian-based methods like TRPO-Lag and PPO-Lag. The connection between cumulative discounted cost and average cost is well-established (Puterman, 2014):

$$\lim_{\gamma \to 1^-} (1 - \gamma) V_C^\pi(s_0, \phi_0) = \xi^\pi(s_0, \phi_0),$$

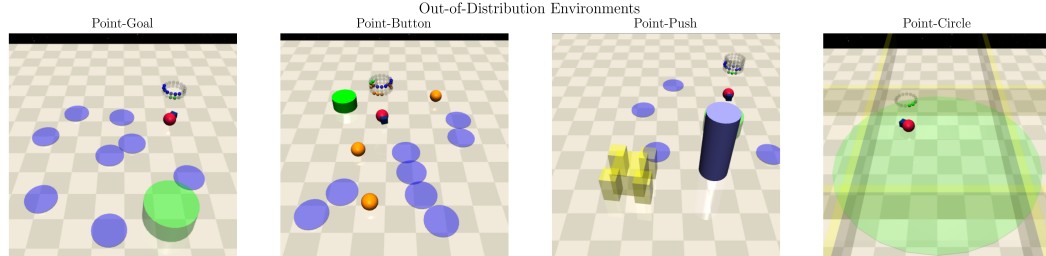

Figure 13: Four out-of-distribution environments for evaluation.

where $V_C^\pi(s_0, \phi_0)$ denotes the value function for the cost under policy $\pi$ starting from state $s$, parameter $\phi$, and $\xi^\pi(s, \phi) = \lim_{H \to \infty} \frac{1}{H} \mathbb{E}_{\pi^*, T_\phi} \left[ \sum_{t=0}^{H-1} C_d(s_t, a_t, s_{t+1}) \mid s_0 = s, \phi \right]$ represents the expected average cost for parameter $\phi$.

## O    EXPERIMENTAL DETAILS

For out-of-distribution (OOD) evaluation, we modify Safety Gymnasium task environments: Goal, Button, and Push, by adding two additional hazard locations to increase complexity. For Circle, we keep the same layout since the wall already blokcs four sides.

To introduce varying hidden parameters, each episode independently samples gravity, damping, mass, inertia, friction multipliers by randomly selecting one of two intervals, $[0.15, 0.3]$ or $[1.7, 2.5]$, with equal probability and uniformly sampling a value from the chosen interval, ensuring diverse environmental conditions. For Circle task, all settings remain the same except for damping, which is sampled from $[1.7, 2.5]$. This adjustment addresses instability in MuJoCo simulator when combined with Circle task, where agents are expected to learn circling behavior. Lower damping factors render the simulator unstable, necessitating this range.

Training is conducted on an Ubuntu 22.04 server using a Slurm job scheduler, which dynamically allocates computational resources. As resource allocations vary across runs, we do not report runtime comparisons for training.

