# OpenReview forum: "Runtime Safety through Adaptive Shielding: From Hidden Parameter Inference to Provable Guarantees"
_ICLR.cc/2026/Conference — ICLR 2026 Conference Desk Rejected Submission_

### Official Review · Reviewer_Wk9s · 2025-10-26

**Soundness:** 3
**Presentation:** 2
**Contribution:** 2
**Rating:** 4
**Confidence:** 4

**Summary:**

This paper develops a generalized safe RL framework. At the core, authors leverages function encoders to encode the set of possible dynamic models for both safety integrated policy optimization and online conformal prediction based deployment.

**Strengths:**

The paper in general is well-written with clear structure, novelty in online action shielding mechanism, and comprehensive experiments. Reviewer will consider raising score if below questions and concerns are addressed.

**Weaknesses:**

1, Following two papers[1,2] are missing in the literature surveys, which also introduce safety guarantee during optimization process. \
2, The theoretical contribution of Proposition 1 is bit vague and trivial since the optimization happens with the assumption that within the zero-violation policy space? Is it possible to come up with some stronger theoretical results like convergence with newly defined safety-augmented Q function? Or performance bound/equivalence with CMDP approach like CPO[3], etc. ? And how does the safety regularized TRPO mentioned in the Appendix B related with the safety augmented optimization approach mentioned in the paper, particularly where is the KL policy constrained terms come from?\
3, The motivation of using function encoder as dynamic representation is not explained clearly in section 4.2 and introduction. Why other options such as latent state space model, ensemble models, etc. cannot work?\
4, Section 4.1 and 4.2 need more clarification for the whole RL training paradigm. \
5, The results in figure 1 do not support authors' argument that "SRO or Shield-only" approaches are less effective than the combined method.\
6, Reviewer would love to see using other dynamic model instead of function encoder in the ablation study section. Since in the shielding part function encoder seems to only be used to infer next state with lipschitz continuous assumptions, which are pretty common in other dynamic models.




[1]: Choi, Jason, et al. "Reinforcement learning for safety-critical control under model uncertainty, using control lyapunov functions and control barrier functions." arXiv preprint arXiv:2004.07584 (2020).\
[2]: Wang, Yixuan, et al. "Enforcing hard constraints with soft barriers: Safe reinforcement learning in unknown stochastic environments." International Conference on Machine Learning. PMLR, 2023.\
[3]:Achiam, Joshua, et al. "Constrained policy optimization." International conference on machine learning. PMLR, 2017.\

**Questions:**

1, In line 243, how is $b_i$ inferred from the transition samples? \
2, In section 4.2, is it trained on-policy or off-policy with replay buffer? Where is the observed transition samples come from? At which stage of the training loop the underlying dynamics are trained? \
3, In line 257, what is the $\nu$ function here? And is the lipschitz constant provided?\
4, Thm 1 seems to build on optimal policy regarding $J_R$ instead of $J_{aug}$?\
5, How is the ood environments implemented under the safety gym environments? By using different obstacles?

---

> ### Author Response · Authors · 2025-11-25
>
> We sincerely thank the reviewer for their thoughtful feedback and appreciate their comments. Below, we respond to the reviewer’s feedback.
>
> **Relation to Control Theory**: Thank you for highlighting these fundamental references. We agree that control-theoretic approaches are a pillar of safety research. *We have expanded our Related Work section to cite [1,2] and added a detailed comparison in Appendix E*. To summarize the distinction discussed in the new appendix:
> 1. Control-Theoretic Methods (CBF/HJB): These ensure safety by constructing a specific barrier certificate $h(s)$ that satisfies a differential invariance condition (e.g., $\dot{h}(s)+\alpha(h(s)) \geq 0$). While powerful, they typically require explicitly learning this valid barrier function or knowing the system dynamics to certify the safe set.
> 2. Our Statistical Shielding: We frame our method as achieving a probabilistic forward invariance. Instead of learning a barrier function $h(s)$ from scratch, we leverage the Lipschitz structure of the safety cost function directly. In this view, our adaptive conformal bound ( $\Gamma_t$ ) acts as a stochastic disturbance margin.
>
> This allows us to guarantee that the next state remains in the safe set with high probability without learning a valid barrier certificate under distributional shifts.
>
> **Clarification on TRPO & KL Constraints**: To clarify the relationship: SRO is the objective function, while TRPO is the optimization algorithm. The KL-divergence constraint comes strictly from the TRPO trust-region formulation (limiting the step size $\delta$ to ensure monotonic improvement) and is independent of whether we optimize standard rewards or our safety-augmented objective. *We added more detail in Appendix B*.
>
> **New Theoretical Connection of SRO to CPO (Appendix B)**: Thank you for suggesting a comparison with CPO. We have derived a new theoretical result (added to Appendix B) that explicitly links our SRO objective to the worst-case safety bounds found in CPO [3].
> We summarize the result as follows;
>  CPO establishes a bound on the cost return of the updated policy:
> $J_C\left(\pi_{k+1}\right) \leq J_C\left(\pi_k\right)+\frac{\sqrt{2 \delta} \gamma}{(1-\gamma)^2} \epsilon_C^{\pi_{k+1}}$
> where $\epsilon_C^{\pi_{k+1}}​$ is the maximum expected cost advantage (Proposition 2 in [3]).
> In our new analysis, we prove that optimizing the SRO objective $\left( J_{\text{aug}}=J_R+\alpha J_{\text{safe}} \right)$ effectively suppresses this error term. Specifically, under some mild assumptions (detailed in Appendix B), we demonstrates that increasing $\alpha$ in our framework rigorously tightens the worst-case safety bound with the term $O\left(\frac{\sqrt{\delta}}{\alpha}\right)$ for a fixed KL-radius $\delta$. Finally, to explicitly clarify the overall training paradigm of SRO, *we have provided a detailed pseudocode algorithm in Appendix D*. Please see Appendix B and D for more details.
>
> **Motivation for Function Encoders**: We selected the Function Encoder because it is explicitly designed to map context pairs $(s,a,s′)$ to a latent representation of the transition dynamics (i.e., performing inference using coefficients of basis functions). In contrast, standard ensembles or Latent State Space Models (SSMs) typically learn a fixed mapping and lack an inherent mechanism to infer hidden environment parameters from context at test time without retraining. Furthermore, the Function Encoder is highly efficient [4]. Ensembles require forward passes through multiple networks, and complex SSMs often require iterative inference. In contrast, the Function Encoder aggregates context into a latent representation via efficient permutation-invariant operations, enabling the rapid querying required for real-time safety shielding.
>
> *We added Appendix G, which compares our approach against three baselines: a Naive Transformer, an Oracle Probabilistic Ensemble (PEM), and an Oracle MLP*.
> 1. Why Oracle Baselines? Since standard PEM and MLP architectures are not designed to infer hidden parameters ϕ from context alone, we provided them with the true hidden parameters $\phi$ as input. This establishes a performance upper bound (i.e., how well a model could perform if it had perfect knowledge of the environment).
> 2. Results: Our Function Encoder matched the prediction error of these Oracle-informed approaches and outperformed the Naive Transformer. This demonstrates that our learned representation effectively captures the necessary environmental information, closing the gap between "inferring" the parameters and "knowing" them.

---

> ### Author Response · Authors · 2025-11-25
>
> **Performance on Training vs. OOD Tasks**: You are correct that during training (Figure 1), the combined method does not strictly outperform the individual components. This is expected because SRO is sufficient when the environment matches the training distribution, and the Shield is sufficient when the policy is already reasonably safe.
>
> However, we respectfully emphasize that the core contribution of this work is robustness to distributional shifts, which is highlighted in Figure 2 (Out-of-Distribution Evaluation).
> 1. SRO Only: Struggle to keep safety in OOD shifts because the policy is static.
> 2. Shield Only: Successfully prevents violations but often less effective than the combined method (SRO + Shield)
> 3. Combined (SRO + Shield): Figure 2 demonstrates that the synergy is critical here. SRO induces a safer policy during training, while the adaptive Shield provides an additional layer to handle unexpected dynamic shifts to keep safety. This combination yields the most stable performance and the most favorable Pareto trade-off in OOD shifts.
>
> **Answer to Question 1**: The representation $b_\phi=\left[b_1, \ldots, b_k\right]$ corresponds to the coefficients that define the transition dynamics $f$ within the Hilbert space spanned by our learned basis functions $\{g_1, \dots, g_k \}$.
>
> The coefficients are obtained by solving the least-squares problem on the transition samples $D=\{\left(s_i, a_i, s_i^{\prime}\right)\}_{i=1}^N$. This operation is permutation invariant and computationally efficient (linear in the number of samples), allowing us to rapidly infer the environment-specific representation b from the context set at runtime. *Please see updated Appendix H and I for more details*.
>
> **Answer to Question 2**: The Function Encoder operates in an online, in-context manner.
> 1. Context Buffer: Instead of a traditional large replay buffer (which mixes data from different dynamics), we use a temporary episode-length buffer.
> 2. Why? Since hidden parameters $\phi$ are sampled at the start of each episode, transitions from previous episodes correspond to different dynamics. Therefore, we only use the history of the current episode $\{(s_t,a_t,s_{t+1})\}$​ as the context to infer the current dynamics.
>
> **Answer to Question 3**. The function serves as a feature extractor that isolates safety-critical state components. In our navigation tasks, for instance, it is a projection mapping that extracts the robot's $(x,y)$ coordinates from the full high-dimensional state vector. Regarding Lipschitz constant, the constant $L$ depends on the specific design of the safety cost function. For example, please refer to Line 792-796, and Appendix M for more detail.
>
> **Answer to Question 4**. Thank you for spotting this. The notation should refer to the policy optimized under the augmented objective, $J_{\text{aug}}$. We corrected this in the updated manuscript.
> However, it is important to note that the safety guarantee in Theorem 1 is mainly derived from the conformal shielding mechanism part, which is valid regardless of the specific objective function used. The role of optimizing $J_{\text{aug}}$​ (via SRO) is to train a base policy incentivizing lower cost violation.
>
> **Answer to Question 5**. We implement Out-of-Distribution (OOD) environments through two distinct mechanisms:
> 1. Parametric Dynamics Shifts (Physics): We shift the range of hidden physical parameters. For example, if the agent is trained on friction/gravity coefficients in the range [5,10], we evaluate it on disjoint ranges (e.g., [3,5]$\cup$[10,15]) that were never sampled during training.
> 2. Environmental Complexity (Obstacles): regarding obstacles, we increase the density (e.g., increasing the number of hazard regions from 6 to 8), but we do not introduce new types of obstacles.
> - Reasoning: Standard Safety Gym agents use fixed-dimensional sensor inputs. Introducing a new class of obstacle would alter the observation dimension (e.g., shifting from dimension 16 to dimension 28), which a standard MLP policy cannot process without retraining.
> - Future Work: As you noted, handling entirely new obstacle types is an exciting direction. This would require replacing the fixed-input policy with a variable-length encoder (e.g., a Transformer) to process arbitrary numbers of object types, which is compatible with our shielding framework but outside the scope of this work.

---

> ### Author Response · Authors · 2025-11-25
>
> Citations below:
>
> [1]: Choi, Jason, et al. "Reinforcement learning for safety-critical control under model uncertainty, using control lyapunov functions and control barrier functions." arXiv preprint arXiv:2004.07584 (2020).
>
> [2]: Wang, Yixuan, et al. "Enforcing hard constraints with soft barriers: Safe reinforcement learning in unknown stochastic environments." International Conference on Machine Learning. PMLR, 2023.
>
> [3]:Achiam, Joshua, et al. "Constrained policy optimization." International conference on machine learning. PMLR, 2017
>
> [4] Tyler Ingebrand, Amy Zhang, and Ufuk Topcu. "Zero-shot reinforcement learning via function encoders". In International Conference on Machine Learning (ICML), 2024.

---

> > ### Comment · Reviewer_Wk9s · 2025-11-27
> >
> > Thanks for authors' effort, and most of my comments and questions have been addressed. I will raised my score accordingly.

---

> > > ### Author Response · Authors · 2025-11-27
> > >
> > > Thank you for revisiting our rebuttal and for your positive update. We sincerely appreciate your constructive feedback, and are grateful for your support.

---

### Official Review · Reviewer_S71N · 2025-10-29

**Soundness:** 3
**Presentation:** 2
**Contribution:** 2
**Rating:** 4
**Confidence:** 4

**Summary:**

This paper presents an adaptive runtime safety framework for reinforcement learning under hidden environment parameters. The approach combines a safety-regularized CMDP objective, a function encoder that infers latent dynamics online, and an adaptive conformal prediction–based shield providing probabilistic safety guarantees. Experiments on Safe-Gym tasks demonstrate reduced safety violations and reasonable transfer to unseen dynamics.

**Strengths:**

- The framework is clearly formulated with theoretical analysis.
- The empirical results are consistent with the theoretical claims.

**Weaknesses:**

- The overall contribution feels incremental, which is a straightforward combination of several techniques, CMDP training, latent parameter inference, and conformal shielding. The theoretical guarantees follow directly from existing conformal prediction results and standard Lipschitz assumptions, offering limited new insight.

- The related work omits an important branch of control-theoretic safety research, such as control barrier function (CBF)–based safe RL, Lyapunov-based safe control, and Hamilton–Jacobi–Bellman (HJB) reachability methods. These frameworks also provide runtime safety filtering or invariant-set guarantees and are widely recognized in both ML and control literature [1,2,3]. A discussion contrasting these methods with the proposed statistical shielding would clarify the distinct contribution.

- Only a small subset of Safety-Gym tasks (Point, Car, Button, Push) is considered. Recent safe-RL works typically evaluate on a wider variety of environments—including Inverted Pendulum, HalfCheetah-Safe, Hopper-Safe, and Walker2d-Safe—to test robustness under dynamic instability. As such, the empirical section feels limited in demonstrating scalability or generality.

[1] Cheng, Yikun, Pan Zhao, and Naira Hovakimyan. "Safe and efficient reinforcement learning using disturbance-observer-based control barrier functions." Learning for Dynamics and Control Conference. PMLR, 2023.

[2] Ganai, Milan, et al. "Iterative reachability estimation for safe reinforcement learning." Advances in Neural Information Processing Systems 36 (2023): 69764-69797.

[3] Wang, Yixuan, et al. "Enforcing hard constraints with soft barriers: Safe reinforcement learning in unknown stochastic environments." International Conference on Machine Learning. PMLR, 2023.

**Questions:**

1. How is conformal calibration maintained over long episodes when hidden parameters drift gradually?

2. Could the authors discuss connections to control-theoretic runtime safety frameworks (e.g., barrier or reachability-based methods)?

---

> ### Author Response · Authors · 2025-11-25
>
> We sincerely thank the reviewer for their thoughtful feedback and appreciate their comments. Below, we respond to the reviewer’s feedback.
>
> **Clarification on Novelty and Contributions**: We respectfully argue that our contribution goes beyond a simple combination of existing techniques. While we leverage established building blocks (function encoder, conformal prediction), the novelty lies in how they are integrated to solve a challenging problem: ensuring safety under distributional shifts without oracle access, where standard CMDP and shielding methods fail.
>
> Specifically, we highlight two non-trivial contributions:
> 1. The Synergy for Zero-Shot Safety: Standard conformal prediction (CP) cannot handle the distributional shifts inherent in RL transfer tasks. By integrating CP with a function encoder, we bridge this gap, allowing the shield to generalize to unseen environments based on latent context. This is not a standard application of CP.
> 2. Novel Safety-Regularized Objective (SRO): As noted in Section 4.1 and Appendix K, SRO includes a specific regularization term designed to train the policy toward low constraint violation, while mitigating sensitivity on hyperparameter $\alpha$, which is a common failure mode in standard Lagrangian methods as shown in Appendix K. *We have revised to provide additional theoretical analysis of SRO connecting to CPO in Appendix B*.
>
> **Relation to Control Theory**: Thank you for highlighting these fundamental references. We agree that control-theoretic approaches are a pillar of safety research. *We have expanded our Related Work section to cite [1,2,3] and added a detailed comparison in Appendix E*. To summarize the distinction discussed in the new appendix:
>
>  - Control-Theoretic Methods (CBF/HJB): These ensure safety by constructing a specific barrier certificate $h(s)$ that satisfies a differential invariance condition (e.g., $\dot{h}(s)+\alpha(h(s)) \geq 0$). While powerful, they typically require explicitly learning this valid barrier function or knowing the system dynamics to certify the safe set.
>
> - Our Statistical Shielding: We frame our method as achieving a probabilistic forward invariance. Instead of learning a barrier function $h(s)$ from scratch, we leverage the Lipschitz structure of the safety cost function directly. In this view, our adaptive conformal bound ( $\Gamma_t$ ) acts as a stochastic disturbance margin.
>
> This allows us to guarantee that the next state remains in the safe set with high probability without learning a valid barrier certificate under distributional shifts.
>
> **Expanded Evaluation**: We appreciate this suggestion. To demonstrate the scalability and generality of our approach on high-dimensional locomotion tasks, *we have added the result on SafetyHalfCheetahVelocity environment to Appendix J*.
> 1. Results & Scalability:
> High-Dimensional Shift: In this experiment, we varied 14 hidden parameters (friction, body segment lengths, and gear ratio) to create significant distributional shifts.
> 2. Performance: Our framework successfully generalized to these shifts, achieving a favorable Pareto front and enhancing safety compared to the oracle-informed baseline.
> 3. Task Complexity: Interestingly, we observed that satisfying safety constraints in HalfCheetah (typically state/velocity limits) was comparatively easier for the shield than the Safety-Gym navigation tasks, which require avoiding external, spatially distributed obstacles. This suggests that our method is not only robust to high-dimensional dynamic shifts but is also effective in complex constraint environments like Safety-Gym.
>
>
> **Clarification on Problem Setting**: hidden parameters are sampled at the start of an episode and remain fixed throughout that episode.
> 1. Inter-Episode vs. Intra-Episode Shift: Our work addresses distributional shifts across episodes. Because the parameters are fixed during the episode, the dynamics remain stationary for that duration.
> 2. Conformal Validity: Since the dynamics do not change within the episode, the exchangeability assumption required for conformal prediction holds with respect to the calibration data (conditioned on the latent context).
> 3. Addressing Time-Varying Drift: You are correct that if parameters drifted gradually during the episode (non-stationary dynamics), standard exchangeability would be violated. Handling such intra-episode drift would require dynamic adaptation mechanisms, such as recursive least squares (as seen in recent works like [4]), which is an exciting direction for future work.

---

> > ### Comment · Reviewer_S71N · 2025-11-28
> >
> > Thanks for the clear clarification. The added high-dimensional experiment is impressive. I will increase the rate to 6.

---

> > > ### Author Response · Authors · 2025-11-28
> > >
> > > Thank you for revisiting our rebuttal and for intending to raise the score to 6. We truly appreciate it.
> > > I just noticed the numerical rating still shows the previous value. If you have a moment to update it, we will be very grateful.

---

> ### Author Response · Authors · 2025-11-25
>
> Citations below:
>
> [1] Cheng, Yikun, Pan Zhao, and Naira Hovakimyan. "Safe and efficient reinforcement learning using disturbance-observer-based control barrier functions." Learning for Dynamics and Control Conference. PMLR, 2023.
>
> [2] Ganai, Milan, et al. "Iterative reachability estimation for safe reinforcement learning." Advances in Neural Information Processing Systems 36 (2023): 69764-69797.
>
> [3] Wang, Yixuan, et al. "Enforcing hard constraints with soft barriers: Safe reinforcement learning in unknown stochastic environments." International Conference on Machine Learning. PMLR, 2023.
>
> [4] William Ward, Sarah Etter, Jesse Quattrociocchi, Christian Ellis, Adam J. Thorpe, and Ufuk Topcu. "Zero to autonomy in real-time: Online adaptation of dynamics in unstructured environments". https://arxiv.org/pdf/2509.12516

---

### Official Review · Reviewer_xCET · 2025-10-31

**Soundness:** 4
**Presentation:** 4
**Contribution:** 3
**Rating:** 8
**Confidence:** 4

**Summary:**

This paper presents safety-augmented learning objective for RL agents in constrained MDP settings. It tackles an even more challenging problem of MDPs with fully unknown, varying/nonstationary dynamics. The paper leverages notions of linear combination of neural network basis functions, conformal prediction to handle uncertainty and generate uncertainty-aware actions, and a shield to ensure the selection of safe actions at run-time.

**Strengths:**

The paper is clear, concisely written and well organized. It combines related or disparate ideas (e.g. function encoders with zero-shot capability, adaptive CP, and "standard-looking" optimization objectives) into safe RL. The concepts are grounded in theory and then come with promising empirical results. In terms of empirical results, I appreciate the fact that the proposed method is handicapped vis a vis baseline methods and access to hidden dynamics information.

**Weaknesses:**

The weaknesses are not technical. The main weakness is around algorithmic descriptions or approaches. The paper does well to describe SRO and Shielding, but then it is unclear how these are implemented in the results section. For example, one could imagine various policy gradient methods with SRO; but which one is used here? There are more details in the appendix but I believe this could be made more clear in the main body. Likewise with the shielding; which policy (what objective, what algorithmic framework, what networks, etc) is/are used to generate actions that are then shielded?

There is one editorial suggestion that will be rolled into a question below regarding function encoders, which is of central importance to the paper (or perhaps does not have to be, given that the main contribution is SRO and shielding, but this is somewhat obfuscated by the repeated mentioned of function encoders).

The discussion around equation (8) appears to be missing some details, for example telling the reader that the Cost function uses an indicator function, the definition of $L_\nu$ and $\Delta_{max}$. Also, the descriptions of $e(), E$ are hand-wavey.

This is putting way too fine a point on things, but the set definition in Proposition 1 refers to something that is technically not defined (I think?). The paper defines $J_R(\pi)$ earlier but not $J_C(\pi)$. In fact the augmented (safety-regularized objective) is formed as a combination of Q values, not a combination of objectives (or "J" functions). Again, not a huge deal but just noting this for posterity.

**Questions:**

1. A conceptual question and an editorial suggestion rolled into one. The notion of inferring hidden parameters online, and the solution of using function encoders, is quite compelling. First, the question: is the number of basis functions / tunable parameters fixed a priori? And how are the basis functions themselves trained, i.e. the $g_j(x)$'s? My assumption is that these are trained offline in a supervised learning style setting, and then the $b_j$ are updated online. If this is the case, is there a concern about the policy (or policies) used to generate the training data? Related to this -- isn't the zero-shot performance still limited to (or by) whatever experience and performance the neural basis functions have?
Editorial suggestion: perhaps the authors could expand on the intuition behind this and/or technical details to make this more self-contained...maybe a 1-pager in the appendix? I realize there is literature on this and it is cited well in the paper; but it is so central to the paper that it might deserve further treatment.

2. The intuition behind upper and lower bounds of $Q^\pi_{safe}$ is interesting and helpful, and this is a very nice metric. Is there concern about the second scenario, i.e. exploration? It makes sense mathematically and even intuitively, but it does not necessarily make sense in terms of safety. Yes, the likelihood of taking such an action is low, but conditioned on such a low probability action, this measure does not (cannot?) tell us anything about the safety of such an action.

3. The design choice in equation (11) is interesting. Why this choice instead of just picking the action with the highest safety score? Or, why not use a different distribution than uniform and bias the sampling towards higher-safety actions? (e.g. using softmax over safe actions or something)

4. After reading further, I believe the descriptions of "Baselines" (line 344 or so) and the bulleted list under it is slightly misleading. Yes, there is a comparison with six baselines, but is it not an assessment of these baselines with and without the proposed approach? That is, it's not like the paper's "Method XYZ" is compared directly to Methods 1 through 6, which is the case with many other papers. If my understanding is correct, this can be fixed with a slight modification of that text on line 344. To hopefully make my source of confusion more explicit, a bit later the text states, "When evaluating our approach on top of each base algorithm". Are you adding the SRO objective (or Shield) on top of all these baselines? My confusion is amplified by the results. SRO is standalone, Shield is standalone, and then there is SRO+Shield. But what is the actual algorithm at work in these? For example, PPO has an algorithmic framework, and then one can describe the models/architectures used to represent policy networks, critics, how they are trained in sequence or in parallel and with what data, and so forth. I am perhaps not making myself clear, but what is "underneath the hood" of SRO and Shield, how are they trained, etc?

---

> ### Author Response · Authors · 2025-11-25
>
> We sincerely thank the reviewer for their thoughtful feedback and appreciate their comments. Below, we respond to the reviewer’s feedback.
>
> **Additional Detail on Function Encoder**: The number of basis functions is fixed a priori and treated as a tunable parameter. Recent work has also shown that it is possible to estimate an appropriate number of basis functions from the dataset [1]. During supervised training, the neural basis functions $\{g_j\}$ are trained such that $\sum_j b_j g_j(x_i) \approx f(x_i)$ for all training points $(x_i, f(x_i))$. The coefficients are obtained by solving the least-squares problem on the data points. *We added more detail in Appendix H and I*.
>
> **Number of basis function on Function Encoder**:  Regarding the prediction capability of the function encoder, using more basis functions generally provides greater representational capacity. You are correct that function encoder follows a supervised learning setup. Hence, performance depends on the dataset quality. *We added additional experiments in Appendix I* to validate two points:
> (i) a larger number of basis functions improves representational flexibility, and (ii) the function encoder depends on the dataset quality.
>
> In Figure 6, we show that using more basis functions typically reduces evaluation loss in the early stage of training, although different configurations eventually converge to nearly the same performance. This indicates that final performance is determined mainly by dataset quality, while a larger number of basis functions primarily increases representational power. Finally, because the function encoder is trained in a supervised manner using a fixed dataset, the collected samples depend on the policy used to generate them. However, we argue that this dependency is inherent to supervised learning frameworks in general and is not specific to our approach.
>
> **Zero-Shot capability**: The zero shot performance of the function encoder is influenced by the quality of the learned neural basis functions, but it is not strictly limited by them. For example, safety at test time is not determined only by the function encoder. Our shielding mechanism provides an additional layer of protection and can handle out of distribution conditions even when the encoder’s prediction is imperfect. This is also reflected in Figure 2. Although SRO with the function encoder representation can struggle in settings such as the Push task, the combined shielding methods remain robust and maintain safety in OOD shifts.
>
> **Second case of SRO**: We agree with the reviewer that in the second scenario the metric propose only the low likelihood of taking such an exploratory action, and does not by itself guarantee that the action is safe. An exploratory action could possibly have a large cost with very low likelihood. Our argument is that the probability mass assigned to such actions is sufficiently small that their overall contribution to the measure is negligible. In addition, if an exploratory action with high cost is taken, the policy update receives a strong penalty on the action. This drives gradient updates that suppress the probability of selecting that action in subsequent epochs. As training continues and the policy becomes safer, the likelihood of these extreme exploratory actions decreases further.
>
> *We also added new theoretical results for SRO in the updated Appendix B*. In particular, equation (30) in the revised manuscript shows that increasing the regularization parameter $\alpha$ leads to a tighter safety bound, which further supports this behavior.
>
> **Action selection design**: Thank you for this thoughtful question. We experimented with always selecting the action that has the highest safety score, but found that this approach restricts exploration and can prevent the policy from reaching an optimal solution, especially early in training. The uniform distribution in equation (11) provides a simple way to preserve exploration among safe actions without collapsing to a single choice too early.
>
> Using alternative distributions, such as a softmax biased toward higher safety scores, is an interesting direction. This could offer a more adaptive balance between safety preference and exploration. We view this as promising future work and appreciate the reviewer for highlighting this possibility.
>
>
> [1] Su Ann Low, Quentin Rommel, Kevin S. Miller, Adam J. Thorpe, and Ufuk Topcu. Function spaces without kernels: Learning compact Hilbert space representations. https://arxiv.org/abs/2509.20605

---

> ### Author Response · Authors · 2025-11-25
>
> **Baseline Clarification**: You are correct that our method is intended as a safety augmentation wrapper applied on top of existing RL algorithms, rather than a standalone algorithm. This is also why we compared against another safety augmentation wrapper, Saute, which uses state augmentation with a cost budget. In our plots, RCPO-Saute and PPO-Saute appear in red because they are also safety augmentation methods. However, they do not perform inference on hidden parameters, so in the comparison, they directly observe the hidden parameters. We apologize for the ambiguity in the original description near line 344. We have revised the text to clearly explain that our approach is built on top of RCPO (without access to hidden parameters) and how the labels SRO, Shield, and SRO + Shield are defined (Section 5.1).
>
> To clarify what is happening “under the hood,” we summarize the structure as follows:
> 1. The Base Algorithm: In the main paper, the underlying RL algorithm is RCPO without oracle information. This was previously mentioned at line 421, and we added additional clarification at line 362.
>
> 2. The Components:
> SRO: Refers to the Base (RCPO w/o oracle) + our Safety Reward Objective.
> Shield: Refers to the Base (RCPO w/o oracle) + our learned Shielding mechanism.
> SRO+Shield: Refers to the Base (RCPO w/o oracle) + both components.
>
> 3. Comparison vs. Baselines: The baselines listed in the paper (such as standard RCPO and PPO-Lag) are separate safe RL algorithms that have access to full oracle information. Our goal is to show that our augmentation allows a blind agent (no oracle) to achieve safety behavior that is competitive with or better than these oracle-informed baselines. We also validated this on PPO-Lag without oracle information in Appendix G, where we observe that the shielding mechanism consistently improves safety across different underlying RL algorithms.

---

> > ### Comment · Reviewer_xCET · 2025-11-26
> >
> > I acknowledge receipt of the rebuttal.
> >
> > I missed an important detail that was already in Appendix G and think many of the revisions are helpful (and subtle enough to not break up the original flow of the paper).

---

> > > ### Author Response · Authors · 2025-11-26
> > >
> > > Thank you for the thoughtful follow-up and for revisiting our rebuttal. We appreciate your recognition of the clarifications and revisions, and we are so grateful for your constructive feedback throughout the review process.

---

### Author Response · Authors · 2025-11-25
**General Comment**

We sincerely thank the reviewers for their constructive and insightful feedback.
All reviewers (xCET, S71N, Wk9s) highlighted the theoretical foundation of the paper as well as the thorough and fair experimental validation.
This work introduces three integrated contributions for safe RL under distribution shifts:
1. A safety-regularized objective (SRO) that proactively enforces constraints throughout training, incentivizing actions inducing lower long term cumulative cost.
2. Online hidden-parameter inference using a function encoder integrated with shielding framework to infer unknown environment parameters from observed transitions.
3. Adaptive shielding, a reactive framework that robustly handles distribution shifts and model uncertainty by integrating a function encoder and conformal prediction with the framework.
Extensive experiments and theoretical analyses confirm that each component improves safety. Individually, both SRO and adaptive shielding guide a policy toward safer behaviors. Together, they provide the most robust performance, especially under out-of-distribution shifts.

Following the reviewers’ thoughtful suggestions, we have further strengthened the paper as follows:

**Analysis and additional experiments on function encoder**: As reviewer xCET noted, we added more detail on the function encoder along with additional experiments to address the reviewer’s question (Appendix H and I) . As reviewer Wk9s requested, we also compared the function encoder representation with a naive transformer based method, an oracle probabilistic ensemble model, and an oracle MLP (Appendix I).

**SRO in relation to control theory and CPO**: As reviewer S71N suggested, we added a dedicated section in Appendix E that explains the differences between safety assurance in control theory and our shielding mechanism. As reviewer Wk9s requested, we also connected SRO to the theoretical foundations of the CPO algorithm in Appendix B.

**Additional experiments**: As reviewer S71N requested, we added a SafetyHalfCheetahVelocity task to demonstrate the generality of our approach. This task requires the agent to maintain a safe velocity while moving forward. As in the navigation tasks, our proposed method improves safety while preserving strong performance (Appendix J).

**Additional clarifications and revisions**: As reviewers S71N and Wk9s noted, we added a discussion of related work in control theory to the introduction. As reviewers xCET and Wk9s suggested, we added pseudo code in Appendix D that illustrates how the safety regularized objective is used during optimization. We also expanded several explanations throughout the paper in response to the reviewers comments.

Again, we sincerely thank all reviewers for their thoughtful and constructive feedback. All concerns have been addressed through new experiments, additional ablations, theoretical analysis, and clarifications in the revised manuscript. **Updates to the main paper are marked in **red** for easy reference.**

---

### Note · Program_Chairs · 2026-01-17
**Submission Desk Rejected by Program Chairs**

The following references in this submission do not refer to real documents and/or have major errors in bibliographic information:

 Wenhao Xiao, Saït Caliskan, and Calin Belta. Safe reinforcement learning via neural control
barrier functions. IEEE Transactions on Neural Networks and Learning Systems, 2023. doi:
10.1109/TNNLS.2023.3288689.